# Pleiotropic effects of trisomy and pharmacologic modulation on structural, functional, molecular, and genetic systems in a Down syndrome mouse model

Sergi Llambrich[1], Birger Tielemans[1], Ellen Saliën[1], Marta Atzori[2], Kaat Wouters[3], Vicky Van Bulck[3], Mark Platt[4], Laure Vanherp[1], Nuria Gallego Fernandez[5], Laura Grau de la Fuente[5], Harish Poptani[4], Lieve Verlinden[6], Uwe Himmelreich[1], Anca Croitor[1], Catia Attanasio[2], Zsuzsanna Callaerts-Vegh[3], Willy Gsell[1], Neus Martínez-Abadías[5]*[†], Greetje Vande Velde[1]*[†]

[1]Biomedical MRI, Department of Imaging and Pathology, KU Leuven, Leuven, Belgium; [2]Department of Human Genetics, KU Leuven, Leuven, Belgium; [3]Laboratory of Biological Psychology, KU Leuven, Leuven, Belgium; [4]Centre for Preclinical Imaging, Department of Molecular and Clinical Cancer Medicine, University of Liverpool, Liverpool, United Kingdom; [5]Departament de Biologia Evolutiva, Ecologia i Ciències Ambientals (BEECA), Facultat de Biologia, Universitat de Barcelona, Barcelona, Spain; [6]Clinical and Experimental Endocrinology, KU Leuven, Leuven, Belgium

*For correspondence:
neusmartinez@ub.edu (NM-A);
greetje.vandevelde@kuleuven.be (GVV)

[†]These authors contributed equally to this work

## Abstract

Down syndrome (DS) is characterized by skeletal and brain structural malformations, cognitive impairment, altered hippocampal metabolite concentration and gene expression imbalance. These alterations were usually investigated separately, and the potential rescuing effects of green tea extracts enriched in epigallocatechin-3-gallate (GTE-EGCG) provided disparate results due to different experimental conditions. We overcame these limitations by conducting the first longitudinal controlled experiment evaluating genotype and GTE-EGCG prenatal chronic treatment effects before and after treatment discontinuation. Our findings revealed that the Ts65Dn mouse model reflected the pleiotropic nature of DS, exhibiting brachycephalic skull, ventriculomegaly, neurodevelopmental delay, hyperactivity, and impaired memory robustness with altered hippocampal metabolite concentration and gene expression. GTE-EGCG treatment modulated most systems simultaneously but did not rescue DS phenotypes. On the contrary, the treatment exacerbated trisomic phenotypes including body weight, tibia microarchitecture, neurodevelopment, adult cognition, and metabolite concentration, not supporting the therapeutic use of GTE-EGCG as a prenatal chronic treatment. Our results highlight the importance of longitudinal experiments assessing the co-modulation of multiple systems throughout development when characterizing preclinical models in complex disorders and evaluating the pleiotropic effects and general safety of pharmacological treatments.

## eLife assessment

This study presents **valuable** findings that examine both how Down syndrome (DS)-related physiological, behavioral, and phenotypic traits track across time, as well as how chronic treatment with

green tea extracts 25 enriched in epigallocatechin-3-gallate (GTE-EGCG), administered in drinking water spanning prenatal through 5 months of age, impacts these measures in wild-type and Ts65Dn mice. The strength of the evidence is **solid**, due to high variability across measures, perhaps in part attributable to a failure to include sex as a factor for measures known to be sexually dimorphic. This study is of interest to scientists interested in Down Syndrome and its treatment, as well as scientists who study disorders that impact multiple organ systems.

## Introduction

Down syndrome (DS) is a developmental disorder with structural, functional, molecular, and genetic alterations that show a dynamic onset and severity (*Antonarakis et al., 2020*; *de Moraes et al., 2008*; *Grieco et al., 2015*; *Keeling et al., 1997*; *LaCombe and Roper, 2020*; *Llambrich et al., 2022a*; *McCarron et al., 2017*; *Steingass et al., 2011*). The alterations associated with DS simultaneously affect multiple systems that are likely interrelated over development, with changes in one system modulating the others (*Llambrich et al., 2022b*). Many studies have provided evidence of these developmental alterations and have demonstrated the ability of dietary supplements such as epigallocatechin-3-gallate (EGCG) or green tea extracts enriched in EGCG (GTE-EGCG) to modulate these systems separately (*Catuara-Solarz et al., 2016*; *de la Torre et al., 2016*; *Goodlett et al., 2020*; *Guedj et al., 2009*; *Llambrich et al., 2022b*; *Rondal, 2020*; *Stagni et al., 2015*; *Starbuck et al., 2021*). However, a holistic evaluation of the simultaneous effects of trisomy and GTE-EGCG in those intertwined systems is missing. The variety in experimental setups and readouts obtained in different preclinical studies hinders the comparison and integration of results and, as a result, evidence is usually contradictory, reporting both positive and negative treatment effects. This lack of consistency can lead to biased interpretations about the etiology of the disorder and the potential effect of pharmacological treatments.

Much of this DS research was based on the Ts65Dn mouse model (*Davisson et al., 1990*; *Reeves et al., 1995*), as it was one of the first preclinical models available for DS and has been widely used for experimental testing. These mice carry a segment with approximately 120 genes homologous to Hsa21 (starting upstream of Mrpl39 to the telomeric end of Mmu16), translocated to a small centromeric part of Mmu17 (*Duchon et al., 2011*; *Herault et al., 2017*; *Reinholdt et al., 2011*). Ts65Dn mice are trisomic for about two-thirds of the genes orthologous to Hsa21, but also carry genes originating from the Mmu17 that are not related with DS, including about 46 protein-coding genes, 35 nonprotein-coding genes and 35 pseudogenes (*Muñiz Moreno et al., 2020*). These genetic alterations do not fully represent DS's aneuploidy and other mouse and rat models have been developed recently that more faithfully represent the trisomic nature of DS (*Kazuki et al., 2020*; *Kazuki et al., 2022*). However, in this study, we used the Ts65Dn mouse model because it recapitulates the main skeletal, brain, cognitive, brain metabolite, and genetic alterations associated with DS (*Blazek et al., 2011*; *Costa et al., 2010*; *Dierssen et al., 2002*; *Escorihuela et al., 1998*; *Gupta et al., 2016*; *Huang et al., 2000*; *Llambrich et al., 2022b*; *Même et al., 2014*; *Starbuck et al., 2021*); and the effects of GTE-EGCG pharmacological modulation have been extensively evaluated using this mouse model (*Catuara-Solarz et al., 2016*; *Goodlett et al., 2020*; *Jamal et al., 2022*; *Llambrich et al., 2022b*; *McElyea et al., 2016*; *Stagni et al., 2016*; *Starbuck et al., 2021*; *Stringer et al., 2017*).

At the structural level, people with DS show skeletal and brain alterations that progress through ontogeny (*Antonarakis et al., 2020*; *de Moraes et al., 2008*; *Fischer-Brandies et al., 1986*; *Keeling et al., 1997*; *LaCombe and Roper, 2020*; *Pearlson et al., 1998*; *Starbuck et al., 2021*). Children with DS have a decreased buildup of bone mass and a low bone turnover rate, resulting in more osteoclast than osteoblast activity, smaller bone area, lower bone mineral density (BMD), and an increased risk of osteoporosis at adulthood (*Carfì et al., 2017*; *Keeling et al., 1997*; *LaCombe and Roper, 2020*). Individuals with DS also present midfacial hypoplasia and flattened nasal bridge along with skull malformations resulting in a shorter, wider, and rounder skull (*Blazek et al., 2011*; *LaCombe and Roper, 2020*; *Richtsmeier et al., 2000*; *Suri et al., 2010*; *Thomas and Roper, 2021*). Previous studies have shown the potential of GTE-EGCG to modulate craniofacial and postcranial morphology, as well as the microarchitecture and BMD of the long bones, showing positive, negative or no treatment effects (*Abeysekera et al., 2016*; *Blazek et al., 2015a*; *Jamal et al., 2022*; *Llambrich et al., 2022a*; *McElyea et al., 2016*; *Starbuck et al., 2021*; *Stringer et al., 2017*). We previously detected dose-, time-, and

anatomical structure-dependent effects in a study using the same mouse model and the same treatment regime with two different GTE-EGCG doses (*Llambrich et al., 2022a*). A dose of 100 mg/kg/day of GTE-EGCG exacerbated facial dysmorphologies (*Starbuck et al., 2021*), modified the skeletal dysmorphologies associated with DS without rescuing the bones shape (*Llambrich et al., 2022a*), and altered the integration between the skull and the brain (*Llambrich et al., 2022b*). However, a lower dose of 30 mg/kg/day significantly reduced the facial dysmorphologies (*Starbuck et al., 2021*) but did not show additional rescuing effects in other skeletal traits (*Llambrich et al., 2022a*).

In DS, the craniofacial size and shape alterations are accompanied with structural brain alterations. People with DS show a reduced overall brain volume from birth, with disproportionately smaller hippocampus and cerebellum, and larger ventricles (*Hamner et al., 2018*; *Movsas et al., 2016*; *Patkee et al., 2020*; *Pinter et al., 2001*; *Rodrigues et al., 2019*; *Smigielska-Kuzia et al., 2011*). Evidence for the effects of GTE-EGCG on brain anatomy is limited. Our own previous study showed that administration of 100 mg/kg/day of GTE-EGCG altered the brain shape of Ts65Dn mice at adulthood (*Llambrich et al., 2022b*), while another study administering green tea polyphenols corresponding to 0.6–1 mg EGCG per day in YACtg152F7 mice showed reduced thalamic-hypothalamic volume and reduced overall brain weight and volume (*Guedj et al., 2009*).

The structural brain alterations in DS are associated with cognitive disabilities, causing functional disability (*Guidi et al., 2011*; *Rodrigues et al., 2019*; *Stagni et al., 2016*). People with DS show relative impairment in executive function, short-term memory, working memory, and explicit long-term memory (*Grieco et al., 2015*; *Lott and Dierssen, 2010*; *McCarron et al., 2017*; *Real de Asua et al., 2015*; *Steingass et al., 2011*; *Zis and Strydom, 2018*); together with a neurodevelopmental delay in the acquisition of both gross and fine motor skills during childhood (*Beqaj et al., 2017*; *Ferreira-Vasques and Lamônica, 2015*; *Frank and Esbensen, 2015*; *Kim et al., 2017*; *Malak et al., 2015*). Regarding cognition, the therapeutic results of green tea extracts are also contradictory. A study administering 60 mg/kg GTE-EGCG to 3- to 4-month-old mBACtgDyrk1a and Ts65Dn mice for 4 weeks rescued working memory (*Souchet et al., 2015*), and interventional studies in humans with DS have shown the ability of (GTE-)EGCG to improve memory recognition, working memory, inhibitory control, and adaptive behavior (*de la Torre et al., 2016*; *De la Torre et al., 2014*). However, other studies found no effect or negative effects of (GTE-)EGCG on cognition. In children aged 6–12 years, a randomized phase Ib clinical trial administering 10 mg/kg/day of EGCG for 6 months combined with a dietary supplement did not improve cognition and functionality (*Cieuta-Walti et al., 2022*). In mice, GTE-EGCG at a dose of 30 mg/kg/day had no effect on visuospatial learning and memory, and even resulted in reduced swimming speed in the Morris water maze and increased thigmotaxic behavior in 5- to 6-month-old Ts65Dn mice (*Catuara-Solarz et al., 2015*).

Several studies have shown that the cognitive functional alterations observed in DS are related with changes at a molecular level on the concentration of hippocampal metabolites, as people with DS show increased myo-inositol levels when compared to euploid population, and people with DS and dementia show reduced levels of N-acetylaspartate (NAA) when compared to people with DS without dementia (*Beacher et al., 2005*; *Lamar et al., 2011*). These findings warrant further investigation into the potential role of hippocampal metabolites in cognitive function, particularly given that few studies to date have investigated the effects of trisomy on the concentration of the main metabolites in the hippocampus of Ts65Dn mice (*Huang et al., 2000*; *Même et al., 2014*; *Santin et al., 2014*) and there is currently no evidence on the effects of (GTE-)EGCG treatment.

Finally, all these structural, functional, and molecular alterations are the result of a complex genetic imbalance involving the triplicated genes in chromosome 21 and the interactions of many genes across the genome (*Olson et al., 2004*). Indeed, skeletal malformations can be associated with the dysregulation of genes such as *Regulator of Calcineurin 1 (RCAN1), Superoxide Dismutase 1 (SOD1), Engrailed Homeobox 2 (EN2), ETS Proto-Oncogene 2 (ETS2), Sonic Hedgehog (SHH), Dual-Specificity Tyrosine-(Y)-Phosphorylation Regulated Kinase 1 A (DYRK1A), SRY-Box Transcription Factor 9 (SOX9), and Orthodenticle Homeobox 2 (OTX2)* (*Arron et al., 2006*; *Billingsley et al., 2013*; *McElyea et al., 2016*; *Roper et al., 2009*; *Thomas and Roper, 2021*; *Weisfeld-Adams et al., 2016*), while cognitive impairment and altered brain development may be associated with *RCAN1, SOD1, Oligodendrocyte Transcription Factor 1 (OLIG1), Oligodendrocyte Transcription Factor2 (OLIG2), SIM BHLH Transcription Factor 2 (SIM2), Down Syndrome Cell Adhesion Molecule (DSCAM), DYRK1A, Down Syndrome Critical Region 1 (DSCR1), Synaptojanin 1 (SYNJ1)* and *Potassium Inwardly Rectifying*

*Channel Subfamily J Member 6 (KCNJ6)* (*Chang et al., 2003*; *Hoeffer et al., 2007*; *Kazemi et al., 2016*; *Kleschevnikov et al., 2017*; *Lana-Elola et al., 2011*; *Stagni and Bartesaghi, 2022*; *Stagni et al., 2018*). From these genes, *DYRK1A*, a gene involved in both skeletal and neuronal development that is overexpressed by trisomy of chromosome 21 (*Blazek et al., 2015a*; *Dierssen and de Lagrán, 2006*), has been proposed as a target gene for therapy (*Atas-Ozcan et al., 2021*; *Becker et al., 2014*; *Dierssen and de Lagrán, 2006*; *García-Cerro et al., 2014*; *Jarhad et al., 2018*; *Ji et al., 2015*; *Lee et al., 2009*; *Rondal, 2020*). The potential of EGCG to modulate gene expression in the Ts65Dn mouse model for DS has limited evidence, with one study administering 200 mg/kg EGCG on embryonic days 7 and 8 twice daily showing decreases in *Protein patched homolog 1* (*Ptch*) and *Ets2* RNA expression and significant increases in *Rcan1* and *Shh* RNA expression in the first pharyngeal arch of Ts65Dn mice at embryonic day (E) 9.5 (*McElyea et al., 2016*).

Since a holistic evaluation of these systems is missing, we designed a longitudinal experimental setup to follow the simultaneous development of structural, functional, molecular, and genetic alterations in the Ts65Dn mouse model. In this study, we conducted a multi-modal in vivo imaging study using micro computed tomography (μCT), magnetic resonance imaging (MRI) and magnetic resonance spectroscopy (MRS) to investigate the integrated development of craniofacial shape, BMD, brain volumes, and hippocampal metabolites in wildtype and Ts65Dn mice. Additionally, we evaluated the changes in body weight and performed a battery of neurodevelopmental and adult cognitive tests to assess cognitive function from birth to adulthood throughout development. At endpoint, we also

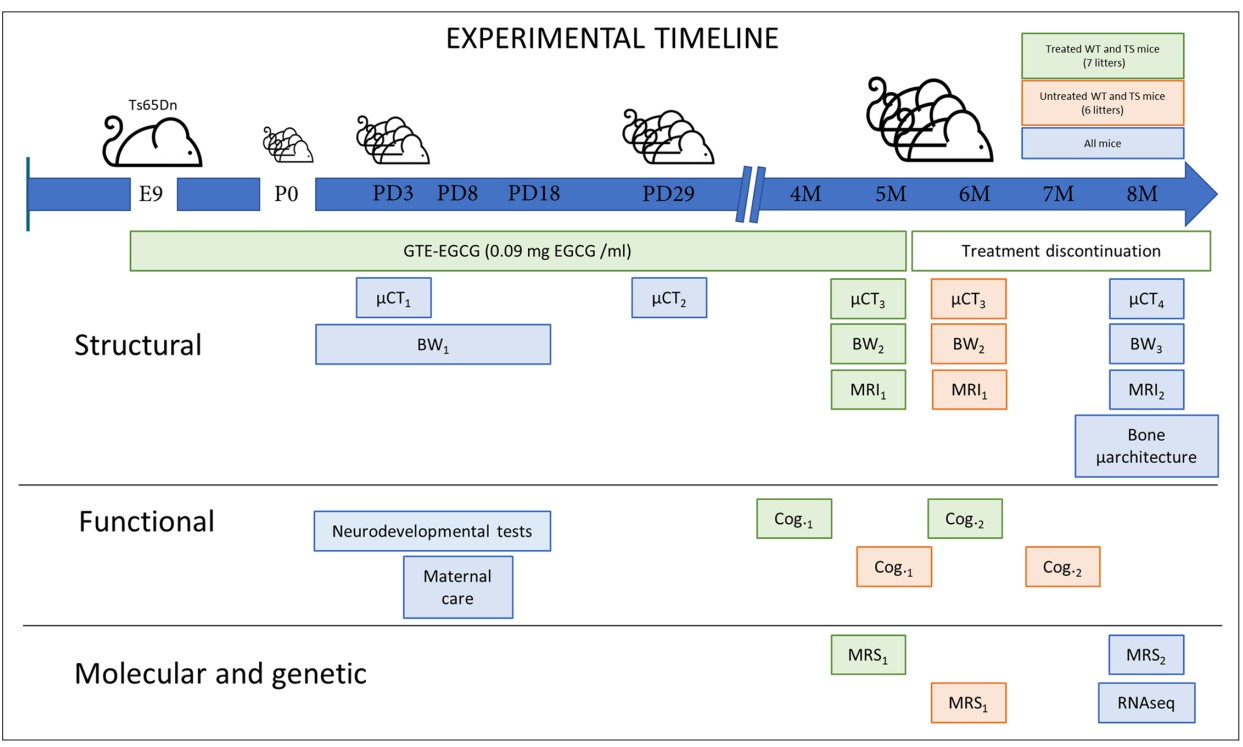

**Figure 1.** Experimental design comparing structural, functional, molecular, and genetic characteristics of wildtype (WT) and Ts65Dn (TS) mice over development and evaluating the effects of a prenatal chronic GTE-EGCG treatment and its discontinuation. Ts65Dn and WT littermates were treated prenatally at a concentration of 0.09 mg EGCG/mL, which corresponds to a dose of 30 mg/kg/day, from embryonic day 9 (**E9**) until mice were 5 months old (5 M). A battery of neurodevelopmental tests was performed daily from postnatal day (PD) 1 to PD18 to evaluate early cognitive development. To monitor maternal care, mice home cages were recorded at PD8 for 24 hr. Mice body weight (BW) was recorded daily from PD1 to PD17 (BW$_1$), and at two additional times before μCT scanning (BW$_2$ and BW$_3$). In vivo micro-computed tomography (μCT) scans were performed four times over development to follow skeletal development: μCT$_1$ at PD3 and μCT$_2$ at PD29 in all mice; μCT$_3$ at 5 M in treated mice, and 6 M in untreated mice; and μCT$_4$ at 8 M in all mice. Additionally, mice were scanned with in vivo magnetic resonance imaging (MRI) and magnetic resonance spectroscopy (MRS) before and after treatment discontinuation to quantify brain volumetric changes and metabolite concentrations in the hippocampal region: MRI$_1$ and MRS$_1$ at 5 M in treated mice and 6 M in untreated mice; and MRI$_2$ and MRS$_2$ at 8 M in all mice. Two batteries of cognitive tests of one-month duration each were performed to evaluate adult cognition: Cog.$_1$ at 4 M in treated mice, and 5 M in untreated mice; and Cog.$_2$ at 6 M in treated mice, and 7 M in untreated mice. At endpoint (8 M), the tibia of all mice was collected to measure its length using a digital caliper and analyze its microarchitecture using ex vivo μCT. At this last stage, cerebellar tissue was also collected to perform RNAseq gene expression analysis.

evaluated tibia microarchitecture from ex vivo μCT scans and used RNAseq to analyze cerebellar gene expression in the same mice at 8 months old (*Figure 1*). Furthermore, we evaluated the pleiotropic effects of prenatal chronic GTE-EGCG treatment and explored the effects of treatment discontinuation on these systems, providing the first controlled longitudinal study assessing the simultaneous effects of treatment across different systems.

## Results

Our longitudinal experimental setup allowed us to investigate the pleiotropic effects of trisomy, GTE-EGCG treatment and treatment discontinuation at the structural, functional, molecular, and genetic levels in the Ts65Dn DS mouse model (*Figure 1*). We compared wildtype (WT) and Ts65Dn (TS) untreated mice to evaluate trisomy effects, and WT and TS treated mice to evaluate treatment effects.

### Structural characterization

Structurally, we investigated the longitudinal effects of trisomy, treatment, and treatment discontinuation on body weight, skeletal system, and adult brain volume (*Figure 2*).

### Body weight

First, we investigated mouse body growth by monitoring body weight over development. TS untreated mice tended to present lower body weights than WT untreated mice over development from birth to adulthood, but these genotype differences did not reach significance at any developmental stage (*Figure 2A*; $P_{BW1}$=0.7630; $P_{BW2}$=0.3688; $P_{BW3}$=0.1094).

Chronic GTE-EGCG treatment, which started prenatally and was maintained until 5 months, significantly reduced the body growth of TS treated mice from PD1 to PD17 (*Figure 2A* at $BW_1$) in comparison with WT untreated (p<0.0001), WT treated (p<0.0001), and TS untreated mice (p<0.0001). At adulthood before treatment discontinuation (*Figure 2A* at $BW_2$), these differences were maintained, but only reached significance when compared to WT untreated (p=0.0322) and WT treated mice (p=0.0103). After three months of treatment discontinuation, both WT and TS treated mice showed a mild increase in body weight, making TS treated mice not significantly different from WT untreated (*Figure 2A*; $P_{BW3}$=0.9284), and causing WT treated mice to be significantly different from WT untreated (*Figure 2A*; $P_{BW3}$=0.0259).

### Skeletal development

Since people with DS show craniofacial abnormalities, reduced BMD, and altered bone microarchitecture (*Carfi et al., 2017*; *Kao et al., 1992*; *LaCombe and Roper, 2020*; *Thomas and Roper, 2021*; *Vicente et al., 2020*), we investigated the effects of genotype, treatment, and treatment discontinuation in these systems by performing in vivo μCT scans throughout development and ex vivo μCT scans at endpoint as indicated in *Figure 1*.

### Craniofacial morphology

We first performed geometric morphometric analyses on the craniofacial 3D models obtained from in vivo μCT scans to investigate craniofacial shape throughout development. The results confirmed our previous findings (*Llambrich et al., 2022a*; *Llambrich et al., 2022b*), showing that TS untreated mice presented a significantly different craniofacial shape than WT untreated mice at $μCT_1$ (p<0.0001) and $μCT_2$ (p=0.0011) (*Supplementary file 2a*). In this study, we further detected that the craniofacial differences induced by genotype persisted until adulthood (*Figure 2B and $P_{μCT3}$*=0.0009; *Figure 2C and $P_{μCT4}$*=0.0055).

Prenatal chronic GTE-EGCG treatment only showed a significant effect in the craniofacial shape at PD3, where both WT and TS treated mice were different from their untreated counterparts ($P_{WT}$=0.0418; $P_{TS}$=0.0005). However, the treatment never rescued the craniofacial shape in TS treated mice, as these mice were significantly different from WT untreated mice at all stages (*Figure 2B and C*; *Supplementary file 2a*). Discontinuing the treatment for three months did not induce any effects (*Figure 2C*), maintaining the craniofacial dysmorphologies already observed at $μCT_3$ (*Figure 2B*).

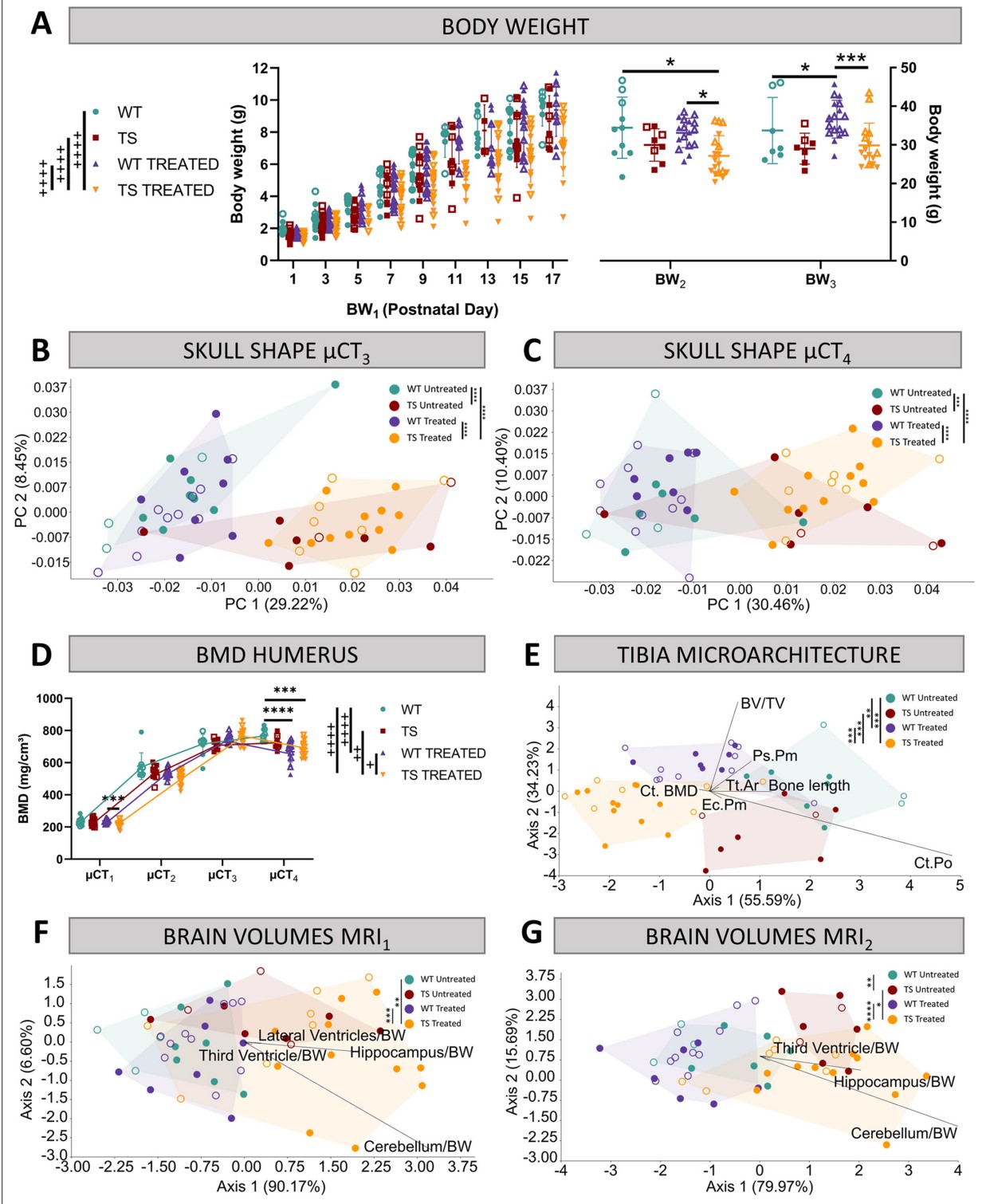

**Figure 2.** Evaluation of trisomy, prenatal chronic GTE-EGCG treatment and treatment discontinuation effects on structural traits. (**A**) Body weight measurements over postnatal development in untreated and treated WT and TS mice. (**B,C**) Skull shape differences between adult WT and TS mice before (**B**) and after GTE-EGCG treatment discontinuation (**C**). Skull shape variation explored by a Principal Component Analysis (PCA) based on the 3D coordinates of landmarks recorded on the surfaces of 3D craniofacial reconstructions from in vivo μCT at μCT$_3$ (**B**) and μCT$_4$ (**C**) timepoints. The landmark configuration for each stage is defined in *Supplementary file 1e*. Scatter plots are presented along with the morphings associated with the negative and positive extremes of the PC1 axis. Statistical differences between groups were assessed by permutation tests based on the Procrustes distances. (**D**) Bone mineral density of the humerus over postnatal development. The in vivo μCT scans of the humerus were used to determine the BMD at μCT$_1$,

*Figure 2 continued on next page*

*Figure 2 continued*

$\mu CT_2$, $\mu CT_3$, and $\mu CT_4$ timepoints. (**E**) Linear discriminant analysis (LDA) based on the results from the tibia microarchitecture tests performed at endpoint (8 M), three months after chronic treatment discontinuation. (**F,G**) LDA based on the brain volumes obtained from in vivo MRI before (**F**) and after GTE-EGCG treatment discontinuation (**G**). The contribution of each variable to separate groups of mice across Axis 1 and Axis 2 is represented in each LDA as lines pointing in the direction of each axis, with longer lines indicating higher contributions. All data are presented as mean +/-standard deviation. (+) p<0.05; (++) p<0.01; (++++) p<0.0001; Mixed-effects analysis across timepoints; (***) p<0.001; (****) p<0.0001; pairwise comparisons. Mice analyzed may differ across stages due to due to uncontrollable technical issues inherent to longitudinal studies, such as scanning failure or mouse death during the experiment but represent overall ontogenetic trajectories. Male mice are indicated with empty symbols. Sample sizes used in each test are provided in *Supplementary file 1a*.

The online version of this article includes the following figure supplement(s) for figure 2:

**Figure supplement 1.** Univariate analysis of tibia microarchitecture in wildtype and trisomic mice at adulthood and effects of prenatal chronic GTE-EGCG treatment and 3 months of discontinuation.

**Figure supplement 2.** Univariate analysis of brain volumes in wildtype and trisomic mice at adulthood and effects of prenatal chronic GTE-EGCG treatment and its discontinuation.

## Bone mineral density

The BMD of the humerus was estimated throughout development from the in vivo μCT scans. Although we observed that TS untreated mice tended to present lower BMD than WT untreated mice over development and adulthood, the differences did not reach significance at any developmental stage (*Figure 2D*, *Supplementary file 2b*).

Prenatal GTE-EGCG chronic treatment significantly modified the developmental trajectory of humerus BMD in WT and TS treated mice ($P_{WT}$ <0.0001; $P_{TS}$=0.0023), but never rescued the trisomic phenotype, as TS treated mice remained different from WT untreated (p<0.0001). At PD3, WT and TS treated mice showed similar levels of BMD as compared to untreated mice (*Figure 2D* at $\mu CT_1$) but tended to show lower BMD at PD29 (*Figure 2D* at $\mu CT_2$). This tendency for reduced BMD was rescued at adulthood (*Figure 2D* at $\mu CT_3$), but after 3 months of treatment discontinuation, both WT and TS treated groups showed a decline in BMD (*Figure 2D* at $\mu CT_4$), which caused WT and TS treated mice to be different than WT untreated control mice ($P_{WT}$=0.0001; $P_{TS}$=0.0002), suggesting a rebound effect following treatment retrieval.

## Tibia development and microarchitecture

Finally, we complemented our structural skeletal analysis by evaluating the length and microarchitecture of the tibia from ex vivo μCT scans at endpoint, as indicated in *Figure 1*.

To obtain a global view of the genotype and treatment effects, we performed a linear discriminant analysis (LDA) combining the results of all tibia microarchitecture tests into a single analysis that maximized the differences among groups of mice (*Figure 2E*). The LDA showed limited overlap between groups, as mice separated by treatment along the first axis, and by genotype along the second axis. These results suggested genotype differences and treatment effects that did not rescue the trisomic phenotype (*Figure 2E*). The multivariate pairwise comparisons after one-way PERMANOVA using Mahalanobis distances detected significant differences between all groups of mice except between WT untreated and TS untreated mice (*Supplementary file 2c*), confirming the treatment effects in both genotypes.

To explore the overall differences detected by the LDA in finer detail, we grouped the results of the tibia microarchitecture tests according to the structural domain they evaluated (cortical bone strength, cortical bone size, and trabecular bone), and repeated the multivariate pairwise comparisons (*Supplementary file 2c*). Even though we did not detect any significant difference between WT untreated and TS untreated mice in any structural domain (*Supplementary file 2c*), TS untreated mice tended to show specific reductions in cortical thickness, polar moment of inertia, cross-sectional area, and bone area (*Figure 2—figure supplement 1* C,E,F,G); and the univariate pairwise comparison tests revealed significant differences in the tibia length (*Figure 2—figure supplement 1A*; p=0.0170) and periosteal perimeter (*Figure 2—figure supplement 1I*; p=0.0154).

After GTE-EGCG chronic treatment and discontinuation, WT treated mice were not significantly different than WT untreated mice for any specific structural domain (*Supplementary file 2c*), but the univariate pairwise tests indicated significantly reduced tibia length (*Figure 2—figure supplement 1A*; p=0.0006) and cortical porosity (*Figure 2—figure supplement 1D*; p=0.0120), together with

increased cortical BMD (*Figure 2—figure supplement 1B*; p=0.0120). TS treated mice were significantly different than TS untreated mice for the cortical bone strength and cortical bone size domains (*Supplementary file 2c*), showing significantly more mineralized (*Figure 2—figure supplement 1B*; p=0.0039), thicker (*Figure 2—figure supplement 1C*; p=0.0400), and less porous (*Figure 2—figure supplement 1D*; p=0.0005) cortical bone. However, they appeared different from WT untreated mice in both cortical domains (*Supplementary file 2c*) as well as in the cortical BMD (*Figure 2—figure supplement 1B*; p=0.0097) and cortical porosity tests (*Figure 2—figure supplement 1D*; p=0.0107). These results could be explained either by adverse effects of the treatment during development that were persistent after discontinuation, or by a negative effect following treatment withdrawal.

## Brain development

Finally, as it was reported that people with DS show altered brain volumes (*Aylward et al., 1997*; *Hamner et al., 2018*; *Movsas et al., 2016*; *Patkee et al., 2020*; *Pinter et al., 2001*; *Rodrigues et al., 2019*; *Smigielska-Kuzia et al., 2011*), we completed our structural evaluation by assessing the volume of the whole brain, along with the volumes of the hippocampal region, the cerebellum and the ventricles using the in vivo MRI scans performed before and after treatment discontinuation (*Figure 1*).

The LDA including the volumes of the subcortical brain regions mentioned above showed genotype separation across Axis 1 at $MRI_1$, which further increased at $MRI_2$, with WT mice occupying the negative extreme of Axis 1 and TS mice occupying the positive extreme (*Figure 2F and G*). The multivariate pairwise comparisons after one-way PERMANOVA using Mahalanobis distances (*Supplementary file 2d and e*) confirmed genotype differences only at $MRI_2$, when WT untreated mice were significantly different than TS untreated (*Figure 2G*; p=0.0065). Contrary to previous evidence in Ts65Dn mice (*Aldridge et al., 2007*; *Duchon et al., 2021*; *Holtzman et al., 1996*; *Insausti et al., 1998*), adult TS untreated mice tended to present larger volumes than WT untreated mice in all brain regions at both $MRI_1$ and $MRI_2$ (*Figure 2—figure supplement 2*). The univariate pairwise tests performed at $MRI_1$ confirmed that TS mice presented a significantly larger brain (*Figure 2—figure supplement 2A*, top; p=0.0066), hippocampal region (*Figure 2—figure supplement 2B*, top; p=0.0254) and whole ventricles volume (*Figure 2—figure supplement 2H*, top; p=0.0427) than WT untreated mice. At $MRI_2$, these differences did no longer reach significance and significant increases emerged in the volumes of the third ventricle (*Figure 2—figure supplement 2E*, bottom; p=0.0479) and fourth ventricle (*Figure 2—figure supplement 2F*, bottom; p=0.0014).

Regarding treatment effects, the LDA at $MRI_1$ showed separation between TS treated mice and the rest of the groups (*Figure 2F*), and the multivariate pairwise comparisons after one-way PERMANOVA confirmed significant differences between WT untreated and TS treated mice (p=0.0047), but not between TS untreated and TS treated mice (p=0.4921). Indeed, TS treated mice tended to show larger volumes than TS untreated mice in all brain regions (*Figure 2—figure supplement 2*), but the univariate pairwise comparison tests only detected a significant difference in the hippocampal region (*Figure 2—figure supplement 2B*, top; p=0.0413). These effects did not rescue the trisomic phenotype, as TS treated mice were significantly different than WT untreated mice in all brain regions (*Figure 2—figure supplement 2*, top).

After 3 months of treatment discontinuation at $MRI_2$, some TS treated mice tended to occupy the same space as WT untreated mice in the LDA (*Figure 2G*), and although the multivariate pairwise comparisons after one-way PERMANOVA did not detect significant differences between WT untreated and TS treated mice (p=0.0609), TS treated mice remained significantly different than WT untreated mice in all univariate brain regions except for the volume of the third ventricle (*Figure 2—figure supplement 2E*, bottom; p=0.0562), indicating limited rescuing effects at this last stage after treatment discontinuation.

## Functional characterization

After evaluating the effects of trisomy and chronic prenatal GTE-EGCG treatment and its discontinuation at a structural level, we investigated its simultaneous effects at a functional level. We performed a battery of cognitive tests to evaluate early neurodevelopment and adult cognition before and after treatment discontinuation (*Figure 3*), at the timepoints specified in *Figure 1*.

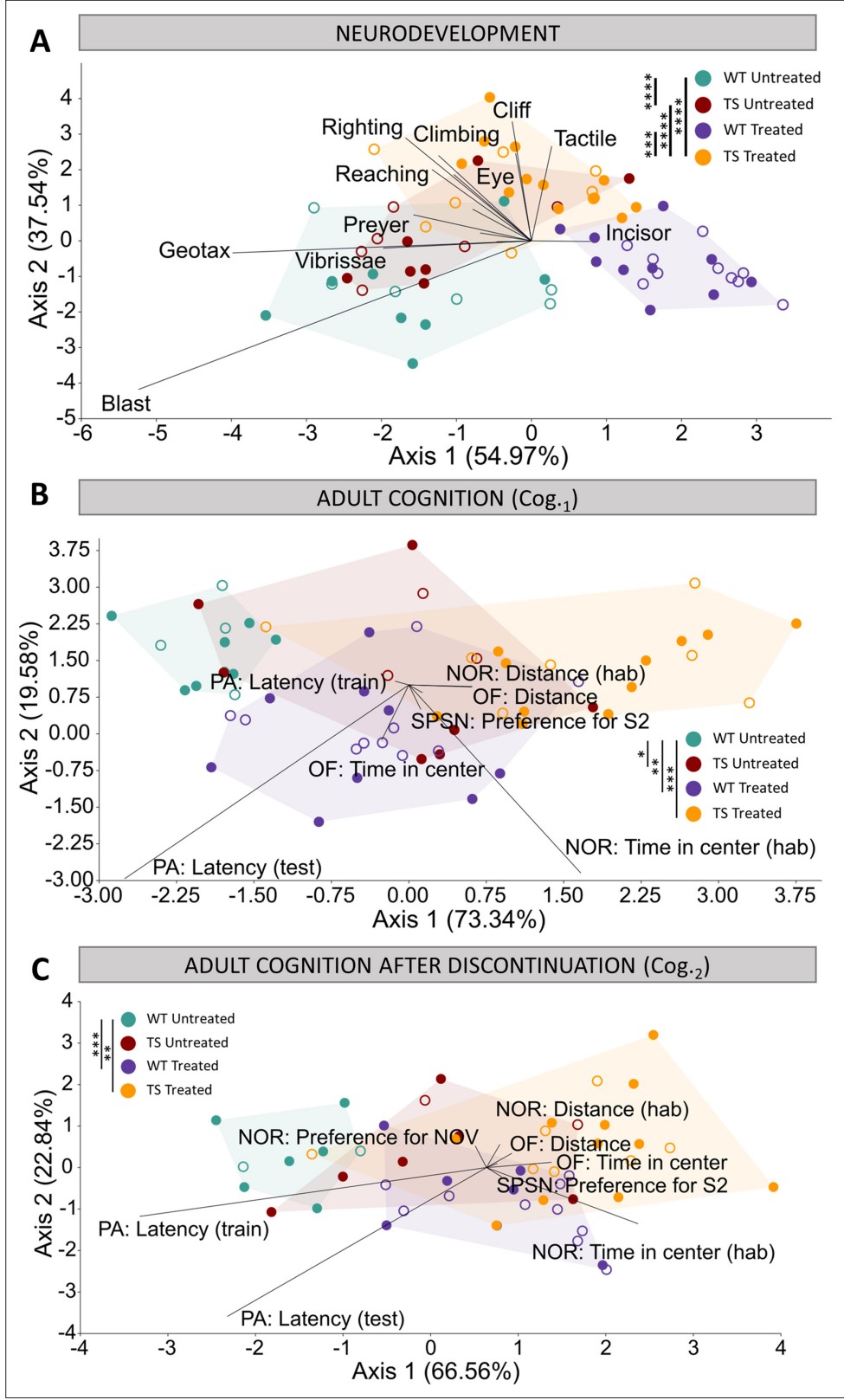

**Figure 3.** Evaluation of trisomy, prenatal chronic GTE-EGCG treatment and treatment discontinuation effects on cognitive function during development and at adulthood before and after treatment discontinuation. (**A**) Linear discriminant analysis (LDA) based on the results from the neurodevelopmental tests performed from PD1 until PD18. (**B,C**) LDA based on the results from the cognitive tests performed at Cog.1 (**B**) and Cog.2 (**C**). The

*Figure 3 continued on next page*

*Figure 3 continued*

contribution of each variable to separate groups of mice across Axis 1 and Axis 2 is represented in each LDA as lines pointing in the direction of each axis, with longer lines indicating higher contributions. Male mice are indicated with empty symbols. Sample sizes for each test are provided in ***Supplementary file 1a***.

The online version of this article includes the following figure supplement(s) for figure 3:

**Figure supplement 1.** Univariate analysis of early postnatal neurodevelopment in WT and TS mice and effects of pre- and postnatal GTE-EGCG treatment.

**Figure supplement 2.** Maternal care evaluation and effects of GTE-EGCG treatment.

**Figure supplement 3.** Univariate analysis of adult cognition in WT and TS mice and effects of prenatal chronic GTE-EGCG treatment and its discontinuation.

## Neurodevelopment

We first evaluated early brain function and neurodevelopment over the first weeks after birth by performing a battery of neurodevelopmental tests from PD1 until PD18 (***Figure 3A***). The LDA with the results from all neurodevelopmental tests combined showed that the treatment was the main differentiating factor among groups of mice, as Axis 1 separated treated and untreated mice (***Figure 3A***). Despite some overlap, Axis 2 separated WT and TS mice. These results suggested mild genotype differences and treatment effects that did not rescue the trisomic phenotype (***Figure 3A***). The multivariate pairwise comparisons after one-way PERMANOVA using Mahalanobis distances (***Supplementary file 2f***) detected significant differences between TS untreated and WT untreated mice (p=0.0001), and between TS treated mice and both TS untreated (p=0.0001) and WT untreated mice (p=0.0001), confirming these findings.

When the neurodevelopmental tests were grouped into functional domains (developmental landmarks, neuromotor tests and reflexes) (***Supplementary file 2f***), the multivariate pairwise comparisons only indicated significant differences between WT untreated and TS untreated mice for the reflexes (p=0.0423). TS untreated mice tended to be delayed in most individual tests (***Figure 3—figure supplement 1***), and the univariate pairwise comparison tests confirmed significant delays in the acquisition rate of the vertical climbing (***Figure 3—figure supplement 1E***; p=0.0256) and tactile response (***Figure 3—figure supplement 1I***; p=0.0064), together with the average day of acquisition of the surface righting response (***Figure 3—figure supplement 1C***; p=0.0320). However, we also observed that TS untreated mice tended to be anticipated in the vibrissae placing, blast response, and Preyer reflex (***Figure 3—figure supplement 1*** J,M,N), even though these results did not reach significance.

Prenatal chronic treatment with GTE-EGCG caused TS treated mice to be significantly different than TS untreated mice for the domains neuromotor tests and reflexes, and WT treated mice to be different from WT untreated mice for the domain reflexes (***Supplementary file 2f***). The univariate tests indicated that treatment significantly modulated the average day of acquisition and acquisition rate of multiple neurodevelopmental tests in both WT and TS treated mice, causing either a delay or an anticipation in most tests (***Figure 3—figure supplement 1***). However, the treatment never rescued the trisomic phenotype, as TS treated mice remained significantly different than WT untreated mice for the three functional domains and most of the individual tests (***Supplementary file 2f***) (***Figure 3—figure supplement 1***).

Finally, we investigated the mothers' behavior during 24 hr at PD8 to evaluate maternal care and observed that, despite being trisomic, all Ts65Dn mothers performed similarly and there was no large intra-group variation for any of the behaviors analyzed, confirming that maternal care did not influence the results of the neurodevelopmental tests (***Figure 3—figure supplement 2***). Furthermore, no significant differences were detected between treated and untreated mothers in any readout (***Figure 3—figure supplement 2***), confirming that the GTE-EGCG treatment did not modulate maternal care.

## Adult cognition

Then, we evaluated adult cognitive performance by means of open field (OF), elevated plus maze (EPM), sociability/preference for social novelty (SPSN), novel object recognition (NOR) and passive avoidance (PA) testing at two timepoints, Cog.$_1$ and Cog.$_2$, before and after treatment discontinuation (***Figure 3B and C***), as specified in ***Figure 1***.

The LDA with the results from all tests combined showed minor overlap between groups at both Cog.$_1$ and Cog.$_2$ (***Figure 3B and C***), with WT untreated mice occupying the negative extreme of Axis 1, WT treated and TS untreated mice presenting an intermediate position, and TS treated mice falling on the positive extreme of Axis 1, opposite to WT untreated mice (***Figure 3B and C***). These results suggested genotype differences between WT and TS untreated mice that were confirmed as significant by the multivariate pairwise comparisons after one-way PERMANOVA at Cog.$_1$ (***Supplementary file 2g***; p=0.0125) but not at Cog.$_2$ (***Supplementary file 2h***; p=0.4596).

When the cognitive tests were grouped into cognitive domains (anxiety, arousal, and memory), the multivariate pairwise comparisons only detected a significant difference between WT untreated and TS untreated mice for the arousal tests at Cog.$_1$ (p=0.0028) (***Supplementary file 2g and h***). Indeed, TS untreated mice showed a tendency to cover longer distances at higher speed in all arousal tests when compared with WT untreated mice (***Figure 3—figure supplement 3*** E-J). These differences reached significance for the distance and speed during OF, as well as for the distance and speed during NOR at Cog.$_1$ (***Figure 3—figure supplement 3*** E,F,I,J, top). TS untreated mice also tended to show increased exploratory and risk-taking behavior, as they usually presented a tendency of increased time spent in the center of the arena during habituation for NOR at Cog.$_1$ (***Figure 3—figure supplement 3C***, top), increased percentage of time spent in the open arm during EPM at Cog.$_1$ (***Figure 3—figure supplement 3D***, top), increased time in the center during OF at Cog.$_2$ (***Figure 3—figure supplement 3A***, bottom), and reduced time to enter the dark chamber during the training session of PA at Cog.$_1$ (***Figure 3—figure supplement 3M***, top), although these differences did not reach statistical significance. Finally, TS untreated mice showed impaired memory robustness and formation at the latest stage, with significantly less preference for the novel object during NOR when compared with WT untreated mice (***Figure 3—figure supplement 3K***, bottom; p=0.0257), and a tendency for reduced latency to enter the dark chamber during training and testing sessions for PA at Cog.$_2$ (***Figure 3—figure supplement 3*** M,N, bottom).

Regarding treatment effects, the results from the LDA at Cog.$_1$ suggested that the prenatal chronic treatment induced changes in both WT and TS treated mice (***Figure 3B***) that exacerbated the differences in TS treated mice and induced cognitive changes in WT treated mice that resulted in a cognitive performance similar to TS untreated mice (***Figure 3B***). Indeed, WT treated mice were significantly different than WT untreated mice when all cognitive tests were considered together as well as for the anxiety and arousal domains (***Supplementary file 2g***). In the case of TS treated mice, the multivariate pairwise comparisons revealed significant differences with WT untreated mice when considering all groups together as well as for the anxiety and arousal domains (***Supplementary file 2g***), confirming the lack of rescuing treatment effects. Both WT and TS treated mice showed significantly increased time in the center during NOR, and increased time in the open arm during EPM at Cog.$_1$ when compared with WT untreated mice (***Figure 3—figure supplement 3*** C,D, top), indicating increased exploratory and risk-taking behavior. Similarly, both WT and TS treated mice showed increased locomotor activity, as they covered significantly more distance at more speed than WT untreated mice for the distance and speed during OF, as well as the distance and speed during NOR at Cog.$_1$ (***Figure 3—figure supplement 3*** E,F,I,J, top). However, in the memory tests (***Figure 3—figure supplement 3*** K-N), the pairwise comparison tests revealed only a significant difference between TS untreated and TS treated mice for the preference for subject 2 (S2) at Cog.$_1$, where treated mice had greater preference for the novel stranger (***Figure 3—figure supplement 3L***, top; p=0.0137).

After treatment discontinuation at Cog.$_2$, the LDA (***Figure 3C***) presented a similar pattern as the LDA at Cog.$_1$ (***Figure 3B***), suggesting that discontinuing the treatment for one month did not substantially change the cognitive patterns induced after chronic GTE-EGCG treatment. However, new significant differences emerged between WT untreated and both WT treated and TS treated mice when evaluating the memory domain (***Supplementary file 2h***), suggesting that the treatment discontinuation altered the memory of both WT and TS mice. Furthermore, TS treated mice remained significantly different than WT untreated mice for all tests combined and all cognitive domains (***Supplementary file 2h***), confirming that the effects of treatment discontinuation did not rescue the trisomic cognitive phenotype. Indeed, significant differences between WT untreated and TS treated mice remained for the percentage of time spent in the open arm during EPM at Cog.$_2$ (***Figure 3—figure supplement 3D***, bottom; p=0.0476), and new significant differences emerged between WT untreated mice and both WT treated and TS treated mice for the time spent in the center and in the periphery during OF

(*Figure 3—figure supplement 3A, B*, bottom), indicating that treated mice still presented a more explorative and risk-taking behavior. A similar trend was observed for the arousal tests, as even though no significant differences were detected between TS treated and TS untreated mice (*Figure 3—figure supplement 3E-J*, bottom), both treated groups showed hyperactivity as the univariate pairwise comparison tests confirmed significant differences between TS treated mice and WT untreated mice for all arousal tests (*Figure 3—figure supplement 3E-J*, bottom); and significant differences were detected between WT treated and WT untreated mice for the distance and speed during SPSN and the distance and speed during NOR (*Figure 3—figure supplement 3G-J*, bottom). In the memory tests, TS treated mice were significantly different than TS untreated (p=0.0046) but not than WT untreated mice for the preference for the novel object during NOR (*Figure 3—figure supplement 3K*, bottom; p=0.3566), which could suggest a beneficial treatment effect. Contrary to the other behavioral tests, the results for PA testing at $Cog._2$ are influenced by the testing at $Cog._1$ as mice may present a robust contextual fear memory trace created during the testing at $Cog._1$, preventing them from entering the dark chamber even in the training phase of the test at $Cog._2$. Indeed, WT untreated mice did not enter the dark box during training at $Cog._2$, while both WT and TS treated mice showed a reduced tendency to enter the dark chamber which reached significance when comparing TS treated mice with WT untreated mice (*Figure 3—figure supplement 3M*, bottom; p<0.0001), indicating less robust contextual memory retention. After re-exposure to the fear conditioning in the training phase of PA at $Cog._2$, all WT mice avoided the dark chamber while a few TS treated mice still showed a reduced tendency to enter the dark chamber (*Figure 3—figure supplement 3N*, bottom), reflecting altered memory robustness.

## Molecular and genetic characterization

Finally, to complement the structural and functional evaluation and to obtain a complete holistic characterization of the alterations associated with DS, we performed in vivo MRS and ex vivo RNAseq at the timepoints specified in *Figure 1* to investigate changes in hippocampal region metabolite concentration and cerebellar gene expression respectively (*Figure 4*).

### Hippocampal region metabolites

We performed in vivo MRS on a voxel placed in the hippocampal region at two timepoints, $MRS_1$ and $MRS_2$, as specified in *Figure 1*.

We first analyzed global differences in the metabolite spectra by performing an LDA containing the relative integrals of 10 spectral regions between 0.0 ppm and 4.3 ppm (*Figure 4*, *Figure 4—figure supplement 1* and *Supplementary file 1b*). The LDA at $MRS_1$ separated WT untreated and TS untreated mice across Axis 1 (*Figure 4A*), and the multivariate pairwise comparisons after one-way PERMANOVA using Mahalanobis distances confirmed significant genotype differences (*Supplementary file 2i*; p=0.0136). After treatment discontinuation at $MRS_2$, the LDA showed milder genotype separation (*Figure 4B*) and the multivariate pairwise comparisons did not detect any significant difference between WT untreated and TS untreated mice (*Supplementary file 2j*; p=0.5790), confirming these findings.

When evaluating the concentrations of specific main metabolites like N-acetylaspartate (NAA), creatine +phosphocreatine (Cr +PCr), choline +phosphocholine + glycerophosphocoline (tCho), myo-inositol (myo-Inos) and taurine (Tau) (*Figure 4—figure supplement 1*), TS untreated mice tended to show lower levels of NAA and tCho at $MRS_1$ when compared with WT untreated mice (*Figure 4—figure supplement 1* A,C), but the univariate pairwise comparison tests only revealed a significant increase in the concentration of Cr +PCr (*Figure 4—figure supplement 1B*, top; p=0.0235). At $MRS_2$, the same pattern was observed but the significant differences between WT untreated and TS untreated mice disappeared (*Figure 4—figure supplement 1*, bottom).

Regarding treatment effects, the LDA at $MRS_1$ separated treated and untreated mice across Axis 2 (*Figure 4A*), suggesting treatment effects that did not rescue the trisomic phenotype. The multivariate pairwise comparisons after one-way PERMANOVA (*Supplementary file 2i*) showed significant differences between TS treated and both WT untreated (p=0.0496) and TS untreated mice (p=0.0361), confirming these findings. When evaluating the specific concentrations of NAA, Cr +PCr, tCho, myo-Inos and Tau, both WT treated and TS treated mice tended to show increased levels of Cr +PCr, which reached significance when comparing TS treated mice with WT untreated mice (*Figure 4—figure*

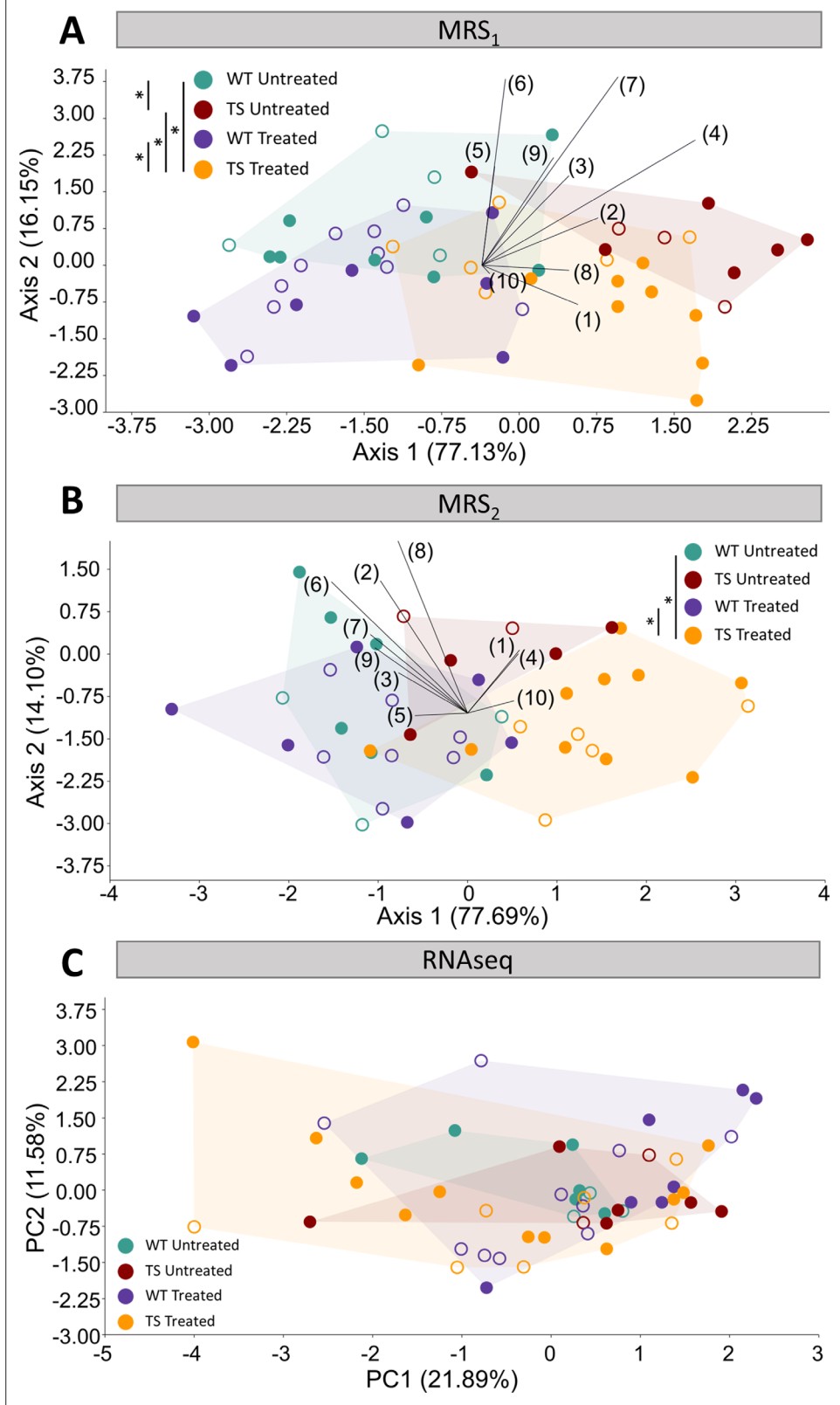

**Figure 4.** Evaluation of trisomy, prenatal chronic GTE-EGCG treatment and treatment discontinuation effects on molecular and gene expression parameters. (**A**,**B**) Linear discriminant analysis (LDA) based on the relative integrals of 10 spectral regions obtained from MRS performed in the hippocampal region at MRS$_1$ (**A**) and MRS$_2$ (**B**). The contribution of each variable to separate groups of mice across Axis 1 and Axis 2 is represented in each

*Figure 4 continued on next page*

*Figure 4 continued*

LDA as lines pointing in the direction of each axis, with longer lines indicating higher contributions. (**C**) Principal component analysis (PCA) based on the normalized expression of the 125 triplicated genes obtained from RNAseq at 8 M. Male mice are indicated with empty symbols. Sample sizes for each test are provided in *Supplementary file 1a*.

The online version of this article includes the following figure supplement(s) for figure 4:

**Figure supplement 1.** Univariate analysis of hippocampal metabolites in wildtype and trisomic mice at adulthood and effects of prenatal chronic GTE-EGCG treatment and its discontinuation.

*supplement 1B*, top; p=0.0317). Conversely, both WT and TS treated mice tended to show reduced levels of myo-inositol and tCho (*Figure 4—figure supplement 1D*, top), causing TS treated mice to have significantly reduced tCho when compared with WT untreated mice (*Figure 4—figure supplement 1C*, top; p=0.0296).

After treatment discontinuation at $MRS_2$, the LDA showed milder treatment separation across Axis 2 (*Figure 4B*) and the multivariate pairwise comparisons did not detect any significant difference between TS untreated and TS treated mice (*Supplementary file 2j*; p=0.2111). However, the differences between TS treated and WT untreated mice remained significant (p=0.0347), indicating that discontinuing the treatment for 3 months did not rescue the trisomic phenotype. When evaluating the specific concentrations of NAA, Cr +PCr, tCho, myo-Inos and Tau, the same pattern was observed and the significant differences between TS treated and WT untreated mice disappeared (*Figure 4—figure supplement 1*, bottom). However, significant differences emerged between WT untreated mice and WT treated mice for the concentration of tCho and myo-inositol (*Figure 4—figure supplement 1* C,D, bottom; $P_{tCho}$=0.0479, $P_{myo-inositol}$=0.0261).

## Gene expression

Finally, since people with DS show impaired motor skills (*Quinzi et al., 2022*), we investigated the effects of genotype and chronic treatment after three months of treatment discontinuation in the cerebellar gene expression by performing RNAseq at endpoint (8 M).

We first performed a PCA using the normalized expression of the 125 genes that are triplicated in the Ts65Dn mouse model and included in our dataset. The PCA showed major overlap between all groups, even though TS treated mice showed a larger range of variation (*Figure 4C*). This result suggested that the differences in gene expression of the triplicated genes were minimal and that GTE-EGCG prenatal chronic treatment and three months of discontinuation did not induce major permanent effects when considering all triplicated genes. The multivariate pairwise comparison tests after one-way PERMANOVA using Euclidean distances did not detect any significant difference between groups, confirming these findings (*Supplementary file 2k*).

Finally, we analyzed the differentially expressed genes (DEGs) between each pair of experimental groups across the entire transcriptome (*Supplementary file 3*). When analyzing the entire transcriptome, we only detected 24 DEGs between WT untreated and TS untreated mice even when applying a loose p-adjusted value of 0.1, indicating small differences in cerebellar gene expression (*Supplementary file 3*). Interestingly, we detected 12 DEGs between WT untreated and TS treated mice, and only 9 of these genes were also found to be differentially expressed between WT untreated and TS untreated mice, which could indicate mild treatment effects in TS mice (*Supplementary file 3*). However, we did not detect any DEGs between TS untreated and TS treated mice, and only five DEGs were detected between WT untreated and WT treated mice, indicating that either the prenatal chronic GTE-EGCG treatment did not largely modulate gene expression or that the treatment had transient effects that were mostly not detected after three months of treatment discontinuation and were not rescuing the trisomic phenotype (*Supplementary file 3*).

## Discussion

We investigated for the first time the simultaneous pleiotropic effects of trisomy and a chronic prenatal and postnatal treatment with GTE-EGCG on several organ systems and levels throughout development in the Ts65Dn mouse model for DS. Our longitudinal and holistic approach overcomes a main limitation of previous research, in which systems were analyzed independently at single time points

using different experimental setups and delivered disparate and contradictory results that could not be generalized. Here, we followed the simultaneous development of structural, functional, molecular, and genetic systems on Ts65Dn mice and provided comparable results of genotype and prenatal chronic GTE-EGCG treatment effects before and after treatment discontinuation.

## Detailed comparison of genotype and GTE-EGCG treatment effects across studies

We here summarized and compared our results with previous studies by domain to discuss whether the Ts65Dn mouse model was representative of DS in the human condition, and whether GTE-EGCG treatment produced any effect that could be translated into clinical practice.

### Structural domain

Starting with body size, we observed that TS mice tended to weigh less than WT mice. However, as in a recent study (*Tallino et al., 2022*), this difference did not achieve statistical significance, and contrasts with previous studies reporting significantly smaller weight of TS mice (*Costa et al., 2010*; *Goodlett et al., 2020*; *Heinen et al., 2012*; *Jamal et al., 2022*; *Roper et al., 2006*). The prenatal chronic GTE-EGCG treatment further reduced the body weight of TS treated mice from PD1 to PD17, but not at adulthood, which is consistent with some studies (*Goodlett et al., 2020*; *Xicota et al., 2020*), but not with others (*Jamal et al., 2022*; *Noll et al., 2022*). Moreover, we observed that WT treated mice showed normal body weight during development, but increased weight after treatment discontinuation, suggesting a possible rebound effect after a prolonged time of treatment administration.

Regarding skeletal morphology, we detected craniofacial alterations that were consistent with previous findings and replicated the human condition (*Llambrich et al., 2022a*; *Llambrich et al., 2022b*; *McElyea et al., 2016*; *Suri et al., 2010*). We also detected modulatory treatment effects on the craniofacial morphology in treated WT and TS mice at PD3, but these did not remain significant at adulthood, which was an unexpected result considering that our previous findings indicated that GTE-EGCG had modulatory effects from birth that were exacerbated later in development at PD29 (*Llambrich et al., 2022a*; *Llambrich et al., 2022b*).

Our multivariate analyses did not detect global tibia microarchitecture alterations associated with genotype and the univariate tests only showed a significant reduction in the length and periosteal perimeter of the proximal tibia, not showing other skeletal alterations previously described (*Abeysekera et al., 2016*; *Blazek et al., 2015a*; *Blazek et al., 2011*; *Blazek et al., 2015a*; *Carfi et al., 2017*; *LaCombe and Roper, 2020*; *Llambrich et al., 2022a*; *Thomas et al., 2020*; *Thomas et al., 2021*). The prenatal chronic treatment and 3 months of discontinuation did not modulate the trabecular bone and caused general but mild adverse effects in the cortical bone of WT and TS mice, not replicating the modulatory effects of green tea polyphenols previously observed on bone microarchitecture (*Blazek et al., 2015b*; *Goodlett et al., 2020*; *Huang et al., 2020*; *Jamal et al., 2022*). Moreover, the GTE-EGCG treatment modified the developmental trajectory of humerus BMD in both WT and TS treated mice with disparate effects throughout development. These results did not match our previous findings (*Llambrich et al., 2022a*) and highlight the variability of treatment responses depending on developmental timing.

In the brain, we detected increased ventricular volumes, which is consistent with the trend observed in our previous study at PD29 (*Llambrich et al., 2022b*) and the ventriculomegaly typically observed in humans and mouse models for DS (*Ishihara et al., 2010*; *Movsas et al., 2016*; *Pearlson et al., 1998*; *Raveau et al., 2017*). However, we also detected increased hippocampal, cerebellar, and whole brain volumes in TS untreated mice that do not match clinical reports indicating reduced brain, hippocampal and cerebellar volumes in humans with DS (*Aylward et al., 1997*; *Hamner et al., 2018*; *Patkee et al., 2020*; *Pearlson et al., 1998*; *Pinter et al., 2001*; *Rodrigues et al., 2019*; *Smigielska-Kuzia et al., 2011*), and other preclinical studies that did not find brain volume differences in Ts65Dn mice (*Aldridge et al., 2007*; *Duchon et al., 2021*; *Holtzman et al., 1996*; *Insausti et al., 1998*). The prenatal chronic treatment did not have major effects, but generally increased the volume of all brain regions in TS treated mice, which is contrary to a previous study indicating that administering a green tea infusion corresponding to 0.6–1 mg EGCG per day from mating until adulthood significantly reduced the brain weight and volume in YACtg152F7 mice (*Guedj et al., 2009*).

## Functional domain

Regarding cognitive function, our results indicated that TS mice showed cognitive delay during early stages of development. As suggested in clinical investigations (*Locatelli et al., 2021*; *Olmos-Serrano et al., 2016b*), the cognitive alterations continued with development, and adult TS mice showed hyperactivity, impaired memory robustness, and a more explorative behavior, in line with previous reports (*Aziz et al., 2018*; *Costa et al., 2010*; *Coussons-Read and Crnic, 1996*; *Dierssen et al., 2002*; *Escorihuela et al., 1995*; *Escorihuela et al., 1998*; *Holtzman et al., 1996*; *Llambrich et al., 2022b*; *Olmos-Serrano et al., 2016b*). Regarding treatment effects, the combined results from all tests suggested that even though the treatment could have had a positive effect reducing the anxiety of treated mice, it likely altered memory robustness and induced a more hyperactive, explorative, and risk-taking behavior. After treatment discontinuation, when the tests were repeated, all groups of mice presented less locomotor activity and explorative behavior. It is not unlikely that this reduction in activity was due to the stress and anxiety induced by the manipulation and testing during the first round of cognitive evaluation. However, TS untreated mice and both WT and TS treated mice still tended to show increased locomotor activity and explorative behavior compared with WT untreated mice, suggesting that the prenatal chronic treatment could have had permanent adverse effects after one month of discontinuation, or could have reduced the anxiety produced by the first round of testing while mice were treated. These results are in line with previous articles indicating no or negative effects of EGCG on cognition in humans and mice (*Cieuta-Walti et al., 2022*; *Goodlett et al., 2020*; *Stringer et al., 2015*; *Stringer et al., 2017*), but are contrary to other articles showing rescuing effects (*Catuara-Solarz et al., 2016*; *de la Torre et al., 2016*; *De la Torre et al., 2014*; *Souchet et al., 2019*; *Stagni et al., 2016*; *Yin et al., 2017*).

## Molecular domain

Regarding hippocampal metabolite concentration, we observed differences between untreated WT and TS mice when evaluating the entire spectra using the LDA, which reflects the analysis of the whole MR spectra rather than selected metabolites, but we only detected a specific significant difference in the levels of Cr +PCr at $MRS_1$, suggesting that the genotype produced mild general molecular differences rather than large specific disruptions in any metabolite. Our results did not reveal any difference in NAA or myo-inositol, which is consistent with a human study showing no differences in the levels of NAA in the hippocampal region of adults with DS without dementia (*Lamar et al., 2011*), but do not replicate the results from another human study indicating increased myo-inositol levels (*Beacher et al., 2005*). Other studies in humans with DS reported decreased NAA levels relative to creatine and myo-inositol in other brain regions, such as the frontal lobes and posterior cingulate cortex (*Lin et al., 2016*; *Smigielska-Kuzia and Sobaniec, 2007*). Studies performed in the Ts65Dn model indicated reduced NAA levels relative to creatine in the hippocampal region (*Huang et al., 2000*; *Même et al., 2014*), which is contrary to our findings, but another study indicated no differences in NAA absolute concentrations (*Santin et al., 2014*). Furthermore, all these mouse studies indicated increased values of inositol or myo-inositol, which we did not detect in this study. These results reflect the large variability of phenotypes associated with DS depending on the developmental stage and region analyzed, as well as the disparity in experimental setups and data analyses between studies, as some studies reported the absolute metabolite concentrations while others reported their relative values normalized to creatine, which could explain the discrepancy among some of the results.

Regarding treatment effects, we detected significant differences between TS treated mice and both TS untreated and WT untreated mice when analyzing the entire spectra using the LDA, but no significant differences between TS untreated and TS treated mice in any specific metabolite. However, TS treated mice presented significantly increased Cr +PCr and reduced tCho levels as compared with WT untreated mice. After treatment discontinuation, these differences were no longer significant, but general differences remained in the LDA between TS treated and WT untreated mice. These results indicated that prenatal chronic GTE-EGCG treatment had mild modulating effects altering the entire hippocampal metabolite profile in TS mice.

## Genetic domain

Regarding cerebellar gene expression, we did not detect large differences in the PCA that compared the expression of triplicated genes across groups of mice. Even though these genes presented an

extra copy number, their expression was not significantly altered, which is consistent with previous articles (*Aït Yahya-Graison et al., 2007*; *Lyle et al., 2004*; *Olmos-Serrano et al., 2016b*; *Saran et al., 2003*).

However, we only detected 24 DEGs between untreated WT and TS mice when analyzing the entire genome, which is a lower number in comparison to other studies (*Aziz et al., 2018*; *Chrast et al., 2000*; *De Toma et al., 2021*; *Olmos-Serrano et al., 2016a*; *Saran et al., 2003*; *Vilardell et al., 2011*). Contrary to previous findings indicating a transcriptome-wide deregulation (*De Toma et al., 2021*; *Letourneau et al., 2014*; *Vilardell et al., 2011*), in our study all the 24 DEGs overexpressed in TS untreated mice were mapped to the mouse chromosomes (Mmu) 16 or 17, except for *Ccdc23,* which mapped to Mmu4. Within these 24 DEGs we detected genes typically related with DS and cognitive disability, such as *App*, *Dyrk1A*, *Dscr3*, *Synj1*, *Ttc3*, *Hmgn1*, *Brwd1*, and *Usp16*; but we did not detect other genes such as *Sod1*, *Sim2*, *Dscam*, *Rcan1*, *Olig1*, *Olig2*, or *Kcnj6* (*Aziz et al., 2018*; *Chrast et al., 2000*; *De Toma et al., 2021*; *Ruparelia et al., 2012*; *Vilardell et al., 2011*). These discrepancies with previous results could be explained by methodological reasons, as previous studies investigated other tissues, pooled data from different mouse models, contained only male mice, investigated different developmental timepoints, or used different RNA quantification techniques.

Even though the prenatal chronic GTE-EGCG treatment used in this study was administered continuously from embryonic day 9 until adulthood, the treatment did not cause any large permanent effects in the gene expression of adult TS mice after 3 months of discontinuation, as we did not detect any DEG between TS untreated and TS treated mice. These results suggest that even though the treatment could have potentially modulated gene expression during a critical window for brain development in TS mice, these effects may have been reverted after treatment withdrawal. In WT mice, five genes were found to be differentially expressed between WT untreated and WT treated mice, of which *Hspa1b* and *Hspa1a* mapped to Mmu17, *Fosl2* mapped to Mmu5, *Actn1* to mapped Mmu12, and *Tm6sf2* mapped to Mmu8; which could indicate permanent genome-wide treatment effects in a few genes in WT mice. Interestingly, from the list of DEGs, only *Hspa1b* was found to interact with *Dyrk1A* (*Rouillard et al., 2016*), suggesting that GTE-EGCG may have other mechanisms of action than DYRK1A kinase activity inhibition.

## The pleiotropic nature of genotype and treatment effects

The combination of multi- and univariate tests performed in our study allowed us to explore in a more comprehensive way the integrated effects of genotype and treatment. Overall, we detected more significant differences when performing multivariate analysis combining the results of different tests together than when we evaluated each test individually, which highlights the pleiotropic nature of DS and GTE-EGCG, as both genotype and treatment caused mild effects in multiple readouts rather than large specific effects in single variables.

For example, we observed that even though TS untreated mice presented mild gene expression alterations in the cerebellum, these mice still showed hyperactivity and increased cerebellar volume. Similarly, TS treated mice presented a larger cerebellum and hyperactivity when compared to WT untreated mice after treatment discontinuation, but only 12 genes were found to be differentially expressed between WT untreated and TS treated mice at that stage. These results can be interpreted as that the altered expression of a few genes was sufficient to alter the brain and cognitive phenotypes, or that the alteration of gene expression during development was no longer detected at adulthood but was sufficient to permanently alter brain volumetry and cognition, which would be in line with previous reports indicating that the altered phenotype in DS was due to small contributions of multiple genes rather than strong effects of a few selected genes (*Antonarakis et al., 2004*; *Chang et al., 2020*; *Chrast et al., 2000*; *De Toma et al., 2021*). Interestingly, trisomy altered the expression of genes related with hypotonia and movement impairment such as *Son*, *Synj1*, *Atp5o*, *Jam2*, and *Paxbp1*; but TS mice still showed increased locomotor activity.

Furthermore, we observed that even though there were no significant differences in the concentration of NAA in the hippocampal region of TS untreated mice, these mice showed altered hippocampal region metabolite spectra, increased hippocampal region volume, and altered memory robustness. Similarly, the treatment did not alter the concentration of NAA in the hippocampal region of TS treated mice, but induced general differences in hippocampal region metabolite spectra, mildly increased hippocampal region volume, and altered cognition. These results suggested that the

cognitive differences induced by both the genotype and treatment were not related with the concentration of NAA in the hippocampal region but rather with general differences in hippocampal region volume and metabolite spectra.

## The advantages of a longitudinal holistic approach

Single holistic and longitudinal experiments evaluating multiple systems simultaneously overcome the limitations of individual analyses and the lack of consistency across studies. The differences between study results highlighted in this study could be due to a large variety of factors, including experimental differences in the mouse model, developmental stage analyzed, sex distribution of the sample, experimental setup, technical differences in data acquisition and analysis, as well as differences in treatment dose, timing, and route of administration. As a result, evidence from different studies is challenging to interpret, and can lead to misinterpretations about the course of the disorder and the effects of pharmacological treatments. With our approach, the differences between experimental setups could be accounted for, analyzing the simultaneous development of different systems in the same mice in a controlled experimental setup, and investigating the related effects of trisomy and treatment in multiple related systems.

Our results support that, overall, the Ts65Dn model reflects the multisystemic nature of DS and recapitulates many characteristics of the trisomic phenotype. As compared to WT mice, Ts65Dn mice presented a trend for reduced body weight over development together with a brachycephalic skull with facial flatness and a trend for reduced BMD in the humerus. These skeletal alterations co-occurred with cognitive delay during early stages of development, hyperactivity, impaired long-term memory, and increased explorative and risk-taking behavior at adulthood. At the molecular and genetic level, Ts65Dn mice presented alterations in the hippocampal metabolite spectra and differential gene expression in the cerebellum. However, our results also revealed phenotypes that did not match the human condition, as Ts65Dn mice did not show altered tibia microarchitecture and showed increased hippocampal, cerebellar, ventricular, and whole brain volumes. Overall, our longitudinal analyses confirm the validity of the model for the structural, functional, molecular and genetic phenotypes associated with DS, but also support the hypothesis of genotypic and phenotypic drift within the Ts65Dn mouse model (*Shaw et al., 2020*), as we generally observed a decrease in the magnitude of alterations induced by the genotype compared to previous articles including our earlier work.

Regarding the treatment effects, our results confirmed that GTE-EGCG modulated most of these systems simultaneously along development. However, our holistic approach revealed that, in general, the treatment did not rescue the trisomic phenotype and even exacerbated some phenotypes over time, such as body weight, tibia microarchitecture, neurodevelopment, adult cognition, and hippocampal metabolite concentration. Although discontinuing the GTE-EGCG administration reduced the treatment effects, it did not rescue the trisomic phenotype, as TS treated mice remained different from WT untreated mice in most systems. Discontinuing the GTE-EGCG treatment for three months only rescued the body weight and brain volume of a few TS mice but increased the weight of WT mice and reduced the BMD of the humerus in both WT and TS mice. Summing up, our preclinical results warn against a chronic GTE-EGCG treatment initiated prenatally and maintained until adulthood with a dosage of 30 mg/kg/day.

## Future directions

Performing specific experiments to evaluate the effects of a potential therapeutic compound on one structure at one timepoint are important first steps to screen for novel therapeutic agents. However, the pleiotropic effects on all involved organ systems over a relevant time course should not be ignored when evaluating its general safety and efficacy, especially in complex disorders like DS.

The experimental pipeline used in this article, from the imaging techniques to the data analysis strategy, are applicable to any rodent model and potential therapeutic compound, allowing the integrated investigation of other therapeutic compounds and DS models that more faithfully replicate the genetic condition of DS, such as the TcHSA21rat and TcMAC21 mouse model (*Kazuki et al., 2020*; *Kazuki et al., 2022*). Furthermore, the simultaneous direct and indirect effects of potential therapeutic agents could be investigated in an integrated manner for other disorders and syndromes with multisystemic alterations such as Apert, Pfeiffer, and Crouzon craniosynostosis syndromes (*Caputo et al., 2016*; *Fernandes et al., 2016*; *Monteagudo, 2020*; *Nopoulos et al., 2007*; *Pirozzi et al., 2018*;

*Roberts et al., 2012*; *Treit et al., 2016*; *Vogels and Fryns, 2006*; *Wilhoit et al., 2017*; *Wozniak et al., 2019*), considering the associated effects in different systems.

With this holistic approach, the preclinical biomedical research field will be able to solve new challenges and answer new questions, understanding the complexities of systemic diseases in a generalized manner and providing a global context to the contributions of specific genes, proteins, molecules, and compounds.

## Materials and methods

### Animals, housing, treatment, and experimental design

Ts65Dn (B6EiC3Sn-a/A-Ts (1716)65Dn) females and B6EiC3Sn.BLiAF1/J males (refs. 005252 and 003647, the Jackson Laboratory Bar Harbor, ME, USA) were obtained from the Jackson Laboratory and crossed within six months to obtain F1 trisomic Ts65Dn (TS) mice and euploid wildtype littermates (WT) that were used throughout the experiment. Mice were housed at the animal facility of KU Leuven in individually ventilated cages (IVC cages, 40 cm long x 25 cm wide x 20 cm high) under a 12 h light/dark schedule in controlled environmental conditions of humidity (50–70%) and temperature (22 ± 2°C) with food and water supplied ad libitum. Date of conception (E0) was determined as the day in which a vaginal plug was present. After birth, all pups were labeled with a non-toxic tattoo ink (Ketchum Animal Tattoo Ink, Green Paste) for identification throughout the longitudinal experiments, as the same mice were used throughout the entire experiment. All procedures complied with all local, national, and European regulations and ARRIVE guidelines and were authorized by the Animal Ethics Committee of KU Leuven (ECD approval number P120/2019).

Mice were genotyped at PD1 by PCR from tail snips adapting the protocol in *Shaw et al., 2020*. Trisomic primers, Chr17fwd-5'-GTGGCAAGAGACTCAAATTCAAC-3' and Chr16rev-5'-TGGCTTAT TATTATCAGGGCATTT-3'; and positive control primers, IMR8545-5'-AAAGTCGCTCTGAGTTGTTAT-3' and IMG8546-5'- GAGCGGGAGAAATGGATATG-3' were used. The following PCR cycle conditions were used: step 1: 94 °C for 2  min; step 2: 94 °C for 30  s; step 3: 55 °C for 45  s; step 4: 72 °C for 1  min (steps 2–4 repeated for 40 cycles); step 5: 72 °C for 7  min, and a 4 °C hold. PCR products were separated on a 1% agarose gel.

We bred a total of 13 litters. Six litters were left untreated and seven litters were treated via the drinking water with GTE-EGCG (Mega Green Tea Extract, Life Extension, USA) at a concentration of 0.09 mg EGCG/mL, as calculated based on the label concentration (45% EGCG per capsule). As EGCG crosses the placental barrier and reaches the embryo (*Chu et al., 2006*), GTE-EGCG treatment started prenatally at embryonic day 9 (E9) via the drinking water of the pregnant dams. After weaning at postnatal day (PD) 21, GTE-EGCG dissolved in water at the same concentration was provided to the young mice ad libitum until 5 months (5 M), when the treatment was discontinued (*Figure 1*). The treatment was prepared freshly every day and water intake was monitored in each cage. The calculated dosage of EGCG received by an adult mouse would be approximately 30 mg/kg/day considering that, on average, early adult mice weigh 20 g and drink 6 mL of water per day according to our measurements. In developing embryos and pups before weaning, the received dosage was lower since previous studies indicate that maternal plasma concentrations of catechins are about 10 times higher than in placenta and 50–100 times higher than in the fetal brain (*Chu et al., 2007*) and EGCG in milk and plasma of PD1 to PD7 pups was detected at low concentrations (*Souchet et al., 2019*).

Mice were allocated to groups according to their genotype and pharmacological intervention: WT and TS mice untreated or treated with GTE-EGCG (*Figure 1*). Investigators were blinded to genotype during animal experimentation, and to genotype and treatment during data analysis. The same mice were longitudinally used throughout the experiment. We estimated the sample size based on behavioral testing. Based on preliminary data, we estimated a standard deviation of 20% and a relevant effect size of 20%. Assuming an alpha level of 0.05 and accepted power of 0.8, we calculated a minimum sample size of 17 animals per condition. Sample sizes varied across groups and developmental stages due to uncontrollable technical issues inherent to longitudinal studies, such as scanning failure or mouse death during the experiment. The litter information containing litter number, treatment administration and sex for each mouse is described in *Supplementary file 1c*. Detailed information regarding sample sizes for each experiment and analysis is provided in *Supplementary file 1a*.

## Structural assessment

### Body weight

Mice body weight was recorded daily from PD1 to PD17 and before each µCT scanning (*Figure 1*).

### Skeletal development

#### In vivo µCT

We performed high-resolution longitudinal in vivo µCT at four timepoints from after birth until adulthood to monitor skeletal development (*Figure 1*). Mice were anesthetized by inhalation of 1.5–2% of isoflurane (Piramal Healthycare, Morpeth, Northumberland, United Kingdom) in pure oxygen and scanned in vivo with the SkyScan 1278 (Bruker Micro-CT, Kontich, Belgium) for 3 min using the optimized parameters specified in *Supplementary file 1d*. In vivo µCT data was reconstructed using a beam hardening correction of 10% (NRecon software, Bruker Micro-CT, Kontich, Belgium).

#### Skull shape analysis

Skull 3D models were automatically generated from reconstructed in vivo µCT scans by creating an isosurface based on specific threshold for bone using Amira 2019.2 (Thermo Fisher Scientific, Waltham, MA, USA). We compared craniofacial morphology in WT and TS mice with and without GTE-EGCG treatment using Geometric Morphometric quantitative shape analyses (*Dryden and Mardia, 1998*; *Hallgrimsson et al., 2015*; *James Rohlf and Marcus, 1993*; *Klingenberg, 2010*). The analysis was based on the 3D coordinates of anatomical homologous landmarks recorded over the skull and face at each developmental stage as described before (*Llambrich et al., 2022a*). The landmark configuration for each stage is defined in *Supplementary file 1e*. Landmarks were acquired using Amira 2019.2.

#### Humerus bone mineral density (BMD)

To calculate humerus BMD from the µCT data, we first computed the humerus mean grey value of each mouse by delimiting a volume of interest of ten slices that was placed right below the deltoid protuberance of the humerus using the CTAn software (Bruker Micro-CT, Kontich, Belgium). Then, we scanned two phantoms with different known densities of hydroxyapatite (100 mg/cm$^3$ and 500 mg/cm$^3$) using the same settings as in the in vivo scans (*Supplementary file 1d*). A calibration line was obtained between the known hydroxyapatite densities and their corresponding grey values. The resulting equation was applied to calculate the BMD of the humerus of each mouse from their mean grey value.

### Tibia length

After sacrifice, the length of the right tibia was measured in all mice using a digital caliper.

### Ex vivo µCT for tibia

The proximal region of the tibia was scanned ex vivo using the Skyscan 1272 high-resolution µCT scanner (Bruker Micro-CT, Kontich, Belgium) with the optimized parameters specified in *Supplementary file 1d*. After scanning the bones, the raw 2D images were reconstructed using NRecon (version 1.7.3.1, Bruker Micro-CT, Kontich, Belgium) and rotated to a standard position using DataViewer (version 1.5.6.2, Bruker Micro-CT, Kontich, Belgium). The reconstructed images were then analyzed using the CTAn software (version 1.17.8.0, Bruker Micro-CT, Kontich, Belgium) as follows.

#### Trabecular analysis

For trabecular bone, a section of 300 slices (1.5 mm) was selected starting 100 slices (0.5 mm) underneath the point in the proximal tibia where the articular condyles met. Then, a region of interest (ROI) was manually defined including the trabecular bone inside the thin cortical outer layer. The descriptions, abbreviations and parameter units are provided in *Supplementary file 1f*.

#### Cortical analysis

For cortical bone, a section of 100 slices (0.5 mm) was selected in CTAn starting 600 slices (3 mm) underneath the reference point in the proximal tibia. The tissue inside the medullary canal was excluded from the ROI. The descriptions, abbreviations and parameter units are provided in *Supplementary file 1g*.

## Brain volume

### In vivo MRI scanning

Mice were also MR scanned in vivo under the same anesthesia (1.5–2% isoflurane, Piramal Healthy-care, Morpeth, Northumberland, United Kingdom) at $MRI_1$ and $MRI_2$ (*Figure 1*) with a 9.4T Bruker Biospec 94/20 small animal µMR scanner (Bruker Biospin, Ettlingen, Germany; 20 cm horizontal bore) equipped with actively shielded gradients (maximum gradient strength 600 mT m$^{-1}$). Axial, coronal, and sagittal images were acquired using a 2D T2 weighted Rapid Acquisition with Relaxation Enhancement (RARE) sequence (repetition time (TR)/ echo time (TE): 3781/33ms; RARE factor: 8; averages: 6; field of view (FOV): 20×20 mm; matrix 128×128; slice number: 35; slice thickness: 0.4 mm; slice gap: 0.1 mm; acquisition time 6 min). A quadrature radiofrequency resonator (inner diameter 7.2 cm, Bruker Biospin) was used for transmission of radiofrequency pulses in combination with and actively decoupled mouse brain surface coil for reception (Bruker Biospin).

### Segmentation of brain regions of interest

After MR image acquisition, brain masks were manually delineated on the axial plane for each mouse using 3D Slicer v5.0.2. (http://www.slicer.org)(*Fedorov et al., 2012*). Then, the masks were fed to the Atlas-Based Imaging Data Analysis (AIDA) pipeline described previously (*Pallast et al., 2019*). In brief, the pipeline consisted of a series of preprocessing steps including skull stripping and bias field correction of the MR images before registration with the Allen Mouse Brain Reference Atlas (*Sunkin et al., 2013*) through a series of affine and non-linear transformations. The volume of the whole brain was extracted from the manually delineated masks and the volumes of the hippocampal region, cerebellum, and ventricles were extracted from the AIDA segmentations. All brain volumes were normalized to body weight to account for the differences in overall body size between WT and TS mice.

## Functional assessment

### Neurobehavioral development

Neurobehavioral developmental tests were carried out daily from PD1 to PD18, as previously described (*Dierssen et al., 2002*; *Llambrich et al., 2022b*). Mothers were separated from their pups before testing. Pups were then taken out one at a time from their home cage for testing, and mothers were returned into the cage after all pups were evaluated. For each neurobehavioral test, we evaluated the acquisition rate and the average day of successful test completion.

In neurodevelopment tests with a presence/absence binary outcome, such as eye opening, pinna detachment, walking, cliff drop aversion, Preyer reflex, blast response, visual placing, reaching response, vibrissae placing and tactile response; the acquisition rate was scored as the percentage of mice that successfully acquired the landmark or response behavior on each day. The day of successful test completion was considered as the day when there was a positive response.

In those neurodevelopment tests measured with categorical non-binary scores, such as incisor eruption, surface righting response, negative geotaxis, vertical climbing, and grasping; the acquisition rate was scored as the daily average score of each group of mice. The day of successful test completion was considered as the day when the highest score was achieved.

### Maternal care

For maternal care monitoring, home IVC cages were transferred to a separate light cycle-controlled room with food and water supplied ad libitum. Cages were videotaped from the top during 24 hr using the Foscam C1 camera with night vision and a transparent Plexiglas cover with holes for ventilation. The recordings were manually inspected every 6 min for 10 s, and maternal behavior was categorized as nursing, pup grooming, digging in nest, eating, drinking, moving, or digging off nest to evaluate the frequency of each maternal behavior.

### Adult cognition

Open Field (OF), Elevated Plus Maze (EPM), Sociability/Preference for Social Novelty (SPSN), Novel Object Recognition (NOR) and Passive Avoidance (PA) tests were performed in this order before and after treatment discontinuation (*Figure 1*).

The OF test was performed in a brightly illuminated Plexiglas arena (50x50 cm) with transparent walls. Dark habituated (30 min) mice were placed in the left bottom corner facing the walls and were left free to explore the arena for 10 min. Movements were recorded using a camera and the tracking software ANY-mazeTM Video Tracking System software (Stoelting Co., IL, USA). Exploration towards the center of the field was considered a readout for reduced anxiety.

The EPM test took place on a plus shaped maze with 2 open and 2 closed arms (5 cmx30 cm) that was elevated 35 cm from the tabletop. Each mouse was placed in the left closed arm, with the snout pointing away from the crossing. After a 1 min habituation time, the trial was initiated manually, letting the mouse spend 10 min in the arena. Four infrared (IR) beams connected to an activity logger recorded the arm entries of the mouse and one beam recorded the percentage of time that the mouse spent in the open arms per minute.

The SPSN set-up consisted of a Plexiglas box (60x15 cm) with three compartments separated with perforated Plexiglas walls. The SPSN test involved three trials. At the first stage of habituation, the test mouse was left to explore the middle chamber for 300 s, while the left and right chambers were empty and visible from the middle chamber. Next, in the Social Preference stage with Subject 1 (S1), the test mouse was placed in the middle chamber for 300 s while one stranger mouse (STR1) was placed in either the left or right chamber, and the other chamber was left empty. Social approach was recorded as time spent close to STR1, and a preference ratio was calculated (Pref=100* Time close to STR1/ (Time close to STR1 +Time close to Empty)). Finally, in the Social Novelty stage with Subject 2 (S2) (300 s), a second stranger mouse (STR2) was placed in the previously empty chamber. Social recognition memory was scored as preference towards STR2 (calculated as ratio: pref=100* (time close to STR2) / (Time close to STR1 +Time close to STR2)). The two stranger mice were C57BL/6 wild-type mice of the same sex as the test mouse and had served before as stranger mice in previous SPSN experiments. Explorative social behavior towards stranger mice was measured using ANY-mazeTM Video Tracking System software (Stoelting Co., IL, USA).

NOR testing started with a habituation phase where the animals were placed during 15 min in a dimly lit open field arena (wooden box 40X40cm painted white). Twenty-four hours later, the animals were reintroduced to the same arena for 10 min with two identical falcon tubes filled with colored liquid that were placed in opposite corners equally distant from the mouse. Exploration time involving sniffing in close proximity (<2 cm) was recorded by an overhead camera and tracking software. Total exploration time was set at minimally 15 s to ensure that object characteristics were encoded. Sixty minutes later, the animals were placed in the arena for 10 min, with one object replaced by a novel object, and exploration time was recorded. Preference for novel object was calculated as ratio (Pref=100* Time novel object / (Time familiar object +novel object)). Objects were randomly assigned as familiar or novel for each mouse, and the left or right position of the novel object was counterbalanced between trials. Exploration time was measured by a camera and ANY-mazeTM Video Tracking System software (Stoelting Co., IL, USA).

The PA experimental set-up consisted of a transparent box illuminated with an LED lamp leading to a dark box with an electrifiable grid connected to a shocker (LE 100–26, Panlab Bioseb, Spain) and a lid. Dark habituated mice were placed in the light box and when they entered the dark box (CS), the latency to enter the dark box was recorded, and a mild foot shock was delivered (US 0.5mA, 2 s, scrambled). The next day, the trial was repeated (without foot shock presentation) and latency to enter the dark compartment was recorded (maximum 300 s). Animals with good memory retention would display a higher latency to enter on day 2. This test was repeated in the same animals after treatment cessation. We noticed that upon re-exposure to the same setup, some animals still remembered the CS-US presentation, and refused to enter the dark box. Therefore, this second testing session reflected the stability of long-term memory.

## Molecular and genetic assessment

### In vivo MRS

MR spectra were acquired as previously reported using a Bruker Biospec 94/20 MR scanner (*Vanherp et al., 2021*; *Weerasekera et al., 2018*). After MRI scanning, MR spectra were acquired from a 2.5×1.25 × 1.5 mm voxel placed in the hippocampal region of the brain using a PRESS sequence with TR/TE2000/20 ms, 320 averages, and localized shimming with no margin. Water suppression was optimized using VAPOR (*Griffey and P. Flamig, 1990*). An unsuppressed water MR spectrum was

acquired before each water-suppressed 1H-MRS spectrum for quantification/referencing. Shimming was performed using FASTMAP, resulting in a final water line width at half height <20 Hz.

For the multivariate analysis of the spectra, the signals were truncated to retain only the region of interest between 0.0 and 4.3ppm. The msbackadj function in Matlab (*The MathWorks Inc, 2022*) was applied for baseline and offset correction. The baseline was estimated within multiple shifted windows of width 150 separation units and extracted from the original signal. The baseline corrected signals were further analyzed by segmenting it into 12 spectral regions, which were integrated and normalized to the total integral using peak integration methodology for metabolite quantification (*Supplementary file 1b*). The two spectral regions that represent contaminations from macromolecules were excluded (number 11 and 12 in *Supplementary file 1b*). Main metabolites present in the respective spectral regions are listed in *Supplementary file 1b*.

For quantification of absolute metabolite concentrations, a similar approach was taken as previously reported (*Weerasekera et al., 2018*). In brief, spectra were processed using jMRUI v6.0 (*Stefan et al., 2009*). Spectra were phase corrected and an HLSVD (Hankel Lanczos Singular Values Decomposition) filter was applied to remove the residual water signal (*Boogaart et al., 1994*). Metabolites were quantified with the QUEST algorithm (*Ratiney et al., 2004*) in jMRUI using a simulated (NMRScopeB) basis set (*Starčuk et al., 2009*). Results were reported in reference to the non-suppressed water signal. A metabolite data base was used as in *Weerasekera et al., 2018*.

### Gene expression

Gene expression analysis was performed on a cerebellar tissue homogenate at endpoint (*Figure 1*). Each cerebellum was dissected and processed independently. Homogenates were obtained with a gentleMACS dissociator (Miltenyi Biotech). Total RNA was then extracted with QIAzol according to the manufacturer's instructions. RNA purity and concentration were assessed by NanoDrop ND-1000 Spectrophotometer and RNA integrity was evaluated by Fragment Analyzer analysis (RIN ≥8). Illumina TruSeq stranded mRNA kit was used for library preparation, samples were pooled and sequenced on a HiSeq4000, single end, 50 bp reads. A minimum of one million reads were obtained per sample. Quality control of raw reads was performed with FastQC v0.11.7 (*Andrews, 2010*). Adapters were filtered with ea-utils fastq-mcf v1.05 (*Aronesty, 2011*). Splice-aware alignment was performed with HISat2 (*Kim et al., 2019*), against the mouse reference genome mm10 using default parameters. Reads mapping to multiple loci in the reference genome were discarded. Resulting Binary Alignment Map (BAM) files were handled with Samtools v1.5 (*Li et al., 2009*). Quantification of reads per gene was performed with HT-seq Count v0.10.0, Python v2.7.14 (*Anders et al., 2015*). Count-based differential expression analysis was performed with R-based (The R Foundation for Statistical Computing, Vienna, Austria) Bioconductor package DESeq2 (*Love et al., 2014*), normalizing absolute counts. Pairwise comparison of the entire genome for all groups was done with default settings and the reported P-values were adjusted for multiple testing with the Benjamini-Hochberg procedure controlling for false discovery rate (FDR; *Supplementary file 3*). Multivariate evaluation of the subset of 125 triplicated genes present in the Ts65Dn mouse model was performed as described below. All mice that survived until endpoint were included in both analyses (*Supplementary file 1a*).

## Statistics

### Univariate evaluation

The developmental trajectories of the body weight, BMD, and acquisition rate of neurodevelopment tests with categorical non-binary scores were longitudinally analyzed by fitting a mixed-effects model as implemented in GraphPad Prism 8.0, using the Geisser-Greenhouse correction and Restricted Maximum Likelihood (REML) fit as described before (*Llambrich et al., 2022b*). The acquisition rate of neurodevelopment tests with a presence/absence binary outcome was longitudinally analyzed using a log-rank test (Mantel-Haenszel approach), considering the day of appearance of the landmark or response as an event using GraphPad Prism 8.0.

We made five pairwise comparisons for all univariate non-longitudinal data: the body weight at adulthood, the BMD at each developmental timepoint, the variables evaluating tibia microarchitecture, the brain volume of the different regions, the average day of acquisition of the neurodevelopmental tests, the variables evaluating adult cognition, and the concentration of the hippocampal metabolites. We compared WT vs. TS untreated mice to evaluate the genotype effect (1), WT untreated vs. WT

treated mice to evaluate the treatment effect in the WT background (2), TS untreated vs. TS treated mice to evaluate the treatment effect in the trisomic background (3), WT untreated vs. TS treated mice to determine whether the treatment had a rescuing effect in trisomic mice (4), and WT treated vs. TS treated mice to evaluate whether the treatment showed different effects in the WT and trisomic background (5). We determined statistical significance for each comparison using univariate statistical tests as previously described (*Llambrich et al., 2022a*). For the BMD, p-values at each developmental stage were adjusted for multiple comparisons using the Benjamini-Hochberg (Q=5%) test.

The results for normality, homoscedasticity and statistical tests performed for each variable can be found in *Supplementary file 1h*. Mice identified as outliers by the ROUT test (*Motulsky and Brown, 2006*) with a Q (maximum desired False Discovery Rate) of 1% were excluded from the analysis. All univariate statistical analysis were performed using GraphPad Prism (v8.02, GraphPad Software, San Diego, California USA).

## Multivariate evaluation

We performed multivariate statistics in all tests with multiple variables: craniofacial shape, tibia microarchitecture, brain volumes, neurodevelopmental tests, adult cognitive tests, brain metabolite concentration and gene expression.

We performed a principal component analysis (PCA) for the craniofacial shape and gene expression analysis. The PCA for the craniofacial shape analysis was based on the 3D coordinates of the set of 27 landmarks recorded to capture craniofacial shape. To extract shape information from the 3D landmark configurations, we performed a Generalized Procrustes Analysis (GPA) followed by a PCA at each stage as described before (*Llambrich et al., 2022b*) using MorphoJ v1.06d (*Klingenberg, 2011*). For the gene expression data, the PCA was based on the rlog normalized expression data of the 125 triplicated genes in the Ts65Dn mouse model according to the MGI-Mouse Genome Informatics Database that were present in our dataset and was performed using PAST v4.1 (*Hammer et al., 2001*).

As we aimed to maximize differences between groups, we performed a linear discriminant analysis (LDA) for the tibia microarchitecture analysis, brain volumetric analysis, neurodevelopmental tests, adult cognitive tests, and hippocampal metabolite concentration analysis using the results obtained from each test. The variables included in each LDA are shown in *Supplementary file 1i*. We performed an LDA for each domain considering genotype+ treatment as the grouping variable using PAST v4.1. As we compared four groups of mice, the LDA created three new axes that were independent among them and explained 100% of variation across groups. In the LDA, the separation among groups was determined by Mahalanobis distances, which account for correlations between standardized variables, and allow to combine measurements with different units in the same analysis.

If slight or no differences were associated with DS or treatment, the mice groups overlapped in the PCA or LDA scatterplot, showing similar phenotypes. If there were differences, the different groups of mice separated from each other.

To statistically quantify differences between WT, TS, WT treated and TS treated mice and answer the five scientific questions formulated above, we performed pairwise permutation tests with 10,000 rounds following the PCAs and LDAs. For the craniofacial shape analysis, we obtained the p-values from the permutation tests based on the Procrustes distances between the average shape of pairs of groups at each developmental stage using MorphoJ v1.06d (*Klingenberg, 2011*). For the gene expression analysis, we obtained the p-values from the pairwise comparisons after a one-way PERMANOVA based on Euclidean distances using PAST v4.1 (*Hammer et al., 2001*). For the tibia microarchitecture tests, brain volumetric tests, neurodevelopmental tests, adult cognitive tests, and hippocampal metabolite concentration tests we obtained the p-values from the pairwise comparisons after a one-way PERMANOVA based on Mahalanobis distances using PAST v4.1 (*Hammer et al., 2001*).

Finally, to better understand the differences between groups, we selected the variables that were testing for a certain domain and grouped them into categories for the tibia microarchitecture tests, neurodevelopmental tests and adult cognitive tests. We then repeated the multivariate pairwise permutation analysis. The categories were cortical bone strength, cortical bone size and trabecular bone for the tibia microarchitecture; developmental landmarks, neuromotor tests and reflexes for the neurodevelopmental tests; and anxiety, arousal, and memory for the adult cognitive tests.

## Acknowledgements

Imaging data was acquired in the Molecular Small Animal Imaging Center (MoSAIC), a core facility of Dept. Imaging and Pathology, Group Biomedical Sciences, KU Leuven. RNAseq was pefomed in the Genomics Core facility of KU Leuven. The authors acknowledge the Laboratory Animal Centre core facility of KU Leuven for support with animal care.

## Additional information

### Funding

| Funder | Grant reference number | Author |
|---|---|---|
| KU Leuven | C24/17/061 | Greetje Vande Velde |
| Marie-Marguerite Delacroix Foundation | Doctoral Fellowship | Sergi Llambrich |

The funders had no role in study design, data collection and interpretation, or the decision to submit the work for publication.

### Author contributions

Sergi Llambrich, Conceptualization, Data curation, Formal analysis, Investigation, Visualization, Writing – original draft, Writing – review and editing; Birger Tielemans, Investigation; Ellen Saliën, Formal analysis, Investigation; Marta Atzori, Kaat Wouters, Vicky Van Bulck, Mark Platt, Laure Vanherp, Nuria Gallego Fernandez, Laura Grau de la Fuente, Anca Croitor, Formal analysis; Harish Poptani, Resources; Lieve Verlinden, Willy Gsell, Methodology; Uwe Himmelreich, Resources, Formal analysis, Methodology; Catia Attanasio, Zsuzsanna Callaerts-Vegh, Resources, Supervision; Neus Martínez-Abadías, Conceptualization, Resources, Data curation, Supervision, Funding acquisition, Visualization, Writing – original draft, Writing – review and editing; Greetje Vande Velde, Conceptualization, Resources, Data curation, Supervision, Funding acquisition, Visualization, Methodology, Writing – original draft, Project administration, Writing – review and editing

### Author ORCIDs

Sergi Llambrich ![ORCID] http://orcid.org/0000-0001-9980-0208
Vicky Van Bulck ![ORCID] http://orcid.org/0000-0002-9355-0567
Uwe Himmelreich ![ORCID] http://orcid.org/0000-0002-2060-8895
Catia Attanasio ![ORCID] http://orcid.org/0000-0001-8077-5719
Zsuzsanna Callaerts-Vegh ![ORCID] http://orcid.org/0000-0001-9091-2078
Willy Gsell ![ORCID] http://orcid.org/0000-0001-7334-6107
Neus Martínez-Abadías ![ORCID] https://orcid.org/0000-0003-3061-2123
Greetje Vande Velde ![ORCID] http://orcid.org/0000-0002-5633-3993

### Ethics

All procedures complied with all local, national, and European regulations and ARRIVE guidelines and were authorized by the Animal Ethics Committee of KU Leuven (ECD approval number P120/2019).

Joint Public Review: https://doi.org/10.7554/eLife.89763.3.sa1
Author Response https://doi.org/10.7554/eLife.89763.3.sa2

## Additional files

### Supplementary files

• Supplementary file 1. Experimental, statistical and methodological details. (a) Sample size for each analysis, experiment, and developmental stage. Differences in sample size between stages are due to technical reasons such as micro-CT or MR scanner not operating on the scanning day, movement scanning artifacts, or mouse death during the experiment. For the tibia microarchitecture, differences in sample size are due to mice detected as outliers. For the MRS, differences in sample

size are due to signal artifacts. For the adult cognitive tests, differences in sample size are due to artifacts during tracking. (b) Table showing the most concentrated metabolites contributing to the respective spectral regions in MRS. ala – alanine, arg – arginine, cre – creatine, cys – cysteine, Hα of AA – α hydrogen of amino acids, ile – iso leucine, leu – leucine, lys – lysine, gaba – γ-amino butyrate, gln – glutamine, glu – glutamane, gly – glycine, GPC – glycerol phosphocholine, his – histidine, NAA – N-acetyl aspartate, PChol – phosphocholine, PCr – phosphocreatine,tyr – tyrosine, val – valine. Figure showing MR spectra and integrated spectral regions for multivariate MRS analysis. The region from 0 to 4.3 ppm of a representative MR spectrum is shown. Regions 11 and 12, representing contaminations from macromolecules, were not included in the analysis as they showed large variability. (c) Litter information. For each mouse, the litter number, genotype, treatment received and sex are provided. (d) μCT scanning parameters used at each stage. (e) Table and figure showing the anatomical definition of craniofacial landmarks. N/A indicates that the landmarks were not acquired at this stage. (A) Set of 34 landmarks characterizing craniofacial shape at $\mu CT_1$ from a 3D reconstruction of a μCT scan. (B) Set of 27 landmarks characterizing craniofacial shape at $\mu CT_2$, $\mu CT_3$ and $\mu CT_4$ from a 3D reconstruction of a μCT scan. (f) Trabecular bone parameters measured from ex vivo micro-CT images at 8 M. (g) Cortical bone parameters measured from ex vivo micro-CT images at 8 M. (h) Normality, homoscedasticity, and statistical tests performed per parameter. If one of the four mice groups was not normally distributed or one pairwise comparison was not homoscedastic, the variable was considered as not normally distributed and/or not homoscedastic.i. Variables included in each Linear Discriminant Analysis or Principal Component Analysis.

• Supplementary file 2. *P*-values resulting from each test. (a) *P*-values resulting from permutation tests (10,000 permutation rounds) based on Procrustes distances among groups for craniofacial shape. Bold font indicates statistically significant values. Pairwise comparisons marked as N/A were not calculated since they did not evaluate any relevant scientific question. (b) *P*-values resulting from the mixed-effects analysis and pairwise tests for humerus BMD. Bold font indicates statistically significant values after Benjamini–Hochberg correction. Pairwise comparisons marked as N/A were not calculated since they did not evaluate any relevant scientific question. (C) *P*-values resulting from the pairwise tests after a one-way PERMANOVA (9,999 permutation rounds) based on Mahalanobis distances for tibia microarchitecture parameters. Bold font indicates statistically significant values. Pairwise comparisons marked as N/A were not calculated since they did not evaluate any relevant scientific question. (d) *P*-values resulting from the pairwise tests after a one-way PERMANOVA (9,999 permutation rounds) based on Mahalanobis distances for brain volumes before treatment discontinuation. Bold font indicates statistically significant values. Pairwise comparisons marked as N/A were not calculated since they did not evaluate any relevant scientific question. (e) *P*-values resulting from the pairwise tests after a one-way PERMANOVA (9,999 permutation rounds) based on Mahalanobis distances for brain volumes after treatment discontinuation. Bold font indicates statistically significant values. Pairwise comparisons marked as N/A were not calculated since they did not evaluate any relevant scientific question. (f) *P*-values resulting from the pairwise tests after a one-way PERMANOVA (9,999 permutation rounds) based on Mahalanobis distances for early neurodevelopmental tests. Bold font indicates statistically significant values. Pairwise comparisons marked as N/A were not calculated since they did not evaluate any relevant scientific question. (g) *P*-values resulting from the pairwise tests after a one-way PERMANOVA (9,999 permutation rounds) based on Mahalanobis distances for adult cognitive tests before treatment discontinuation at $Cog._1$. Bold font indicates statistically significant values. Pairwise comparisons marked as N/A were not calculated since they did not evaluate any relevant scientific question. (h) *P*-values resulting from the pairwise tests after a one-way PERMANOVA (9,999 permutation rounds) based on Mahalanobis distances for adult cognitive tests after treatment discontinuation at $Cog._2$. Bold font indicates statistically significant values. Pairwise comparisons marked as N/A were not calculated since they did not evaluate any relevant scientific question. (i) *P*-values resulting from the pairwise tests after a one-way PERMANOVA (9,999 permutation rounds) based on Mahalanobis distances for MRS spectra before treatment discontinuation. Bold font indicates statistically significant values. Pairwise comparisons marked as N/A were not calculated since they did not evaluate any relevant scientific question. (j) *P*-values resulting from the pairwise tests after a one-way PERMANOVA (9,999 permutation rounds) based on Mahalanobis distances for MRS spectra after treatment discontinuation. Bold font indicates statistically significant values. Pairwise comparisons marked as N/A were not calculated since they did not evaluate any relevant scientific question. (k) *P*-values resulting from the pairwise tests after a one-way PERMANOVA (9,999 permutation rounds) based on Euclidean distances for normalized gene expression data at endpoint. Bold font indicates statistically significant values. Pairwise comparisons marked as N/A were not calculated since they did not evaluate any relevant scientific question.

• Supplementary file 3. Differentially expressed genes for each pairwise comparison.
• MDAR checklist

## Data availability

The data supporting the findings of this study are available in Dryad at https://doi.org/10.5061/dryad.1rn8pk11r.

The following dataset was generated:

| Author(s) | Year | Dataset title | Dataset URL | Database and Identifier |
|---|---|---|---|---|
| Llambrich S, Tielemans B, Saliën E, Atzori M, Wouters K, Bulck Van, Platt M, Vanherp L, Gallego Fernandez N, Grau de la Fuente L, Poptani H, Verlinden L, Himmelreich U, Croitor A, Attanasio C, Callaerts-Vegh Z, Gsell W, Martínez-Abadías N, Vande Velde G | 2024 | Pleiotropic effects of trisomy and pharmacologic modulation on structural, functional, molecular, and genetic systems in a Down syndrome mouse model | https://doi.org/10.5061/dryad.1rn8pk11r | Dryad Digital Repository, 10.5061/dryad.1rn8pk11r |

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
