## [Editor Report · eLife assessment]

This study presents **valuable** findings that examine both how Down syndrome (DS)-related physiological, behavioral, and phenotypic traits track across time, as well as how chronic treatment with green tea extracts 25 enriched in epigallocatechin-3-gallate (GTE-EGCG), administered in drinking water spanning prenatal through 5 months of age, impacts these measures in wild-type and Ts65Dn mice. The strength of the evidence is **solid**, due to high variability across measures, perhaps in part attributable to a failure to include sex as a factor for measures known to be sexually dimorphic. This study is of interest to scientists interested in Down Syndrome and its treatment, as well as scientists who study disorders that impact multiple organ systems.

---

## [Referee Report · Joint Public Review]

Using Ts65Dn - the most commonly used mouse model of Down syndrome (DS) - the goal of this study is two-pronged: (1) to conduct a thorough assessment of DS-related genotypic, physiological, behavioral, and phenotypic measures in a longitudinal manner; and (2) to measure the effects of chronic GTE-EGCG on these measures in the Ts65Dn mouse model. Corroborating results from several previous studies on Ts65Dn mice, findings of this study show confirm the Ts65Dn mouse model exhibits the suite of traits associated with DS. The findings also suggest that the mouse model might have experienced drift, given the milder phenotypes than those reported by earlier studies. Results of the GTE-EGCG treatment do not support its therapeutic use and instead show that the treatment exacerbated certain DS-related phenotypes.

Strengths:

The authors performed a rigorous assessment of treatment and examined treatment and genotypic alterations at multiple time points during growth and aging. Detailed analysis shows differences in genotype during aging as well as genotype with treatment. This study is solid in the overarching methodological approach (with the exception of RNAseq, described below). The biggest strength of the study is its approach and dataset, which corroborate results from a multitude of past studies on Ts65Dn mice, albeit on adult specimens.

Comments on revised submission:

The authors have made numerous changes to address the concerns of the reviewers. The strengths remain: a large, longitudinal data set for the Ts65Dn mouse model across multiple organ systems. The results also clearly show the impact of GTE-EGCG treatment and do not support its therapeutic use.

The authors should report their a priori power calculations that they used when designing their experiment. This should be added to either the Animals or Statistics subsections of the Methods.

---

## [Author Response]

The following is the authors’ response to the original reviews.

**eLife assessment**

This study presents valuable findings that examine both how Down syndrome (DS)-related physiological, behavioral, and phenotypic traits track across time, as well as how chronic treatment with green tea extracts 25 enriched in epigallocatechin-3-gallate (GTE-EGCG), administered in drinking water spanning prenatal through 5 months of age, impacts these measures in wild-type and Ts65Dn mice. However, the strength of the evidence is incomplete, due to high variability across measures, perhaps attributable to a failure to include sex as a factor for measures known to be sexually dimorphic. This study is of interest to scientists interested in Down Syndrome and its treatment, as well as scientists who study disorders that impact multiple organ systems.

**Public Reviews:**
Using Ts65Dn - the most commonly used mouse model of Down syndrome (DS) - the goal of this study is two-pronged: (1) to conduct a thorough assessment of DS-related genotypic, physiological, behavioral, and phenotypic measures in a longitudinal manner; and (2) to measure the effects of chronic GTE-EGCG on these measures in the Ts65Dn mouse model. Corroborating results from several previous studies on Ts65Dn mice, findings of this study show confirm the Ts65Dn mouse model exhibits the suite of traits associated with DS. The findings also suggest that the mouse model might have experienced drift, given the milder phenotypes than those reported by earlier studies. Results of the GTE-EGCG treatment do not support its therapeutic use and instead show that the treatment exacerbated certain DS-related phenotypes.Strengths:The authors performed a rigorous assessment of treatment and examined treatment and genotypic alterations at multiple time points during growth and aging. Detailed analysis shows differences in genotype during aging as well as genotype with treatment. This study is solid in the overarching methodological approach (with the exception of RNAseq, described below). The biggest strength of the study is its approach and dataset, which corroborate results from a multitude of past studies on Ts65Dn mice, albeit on adult specimens. It would be beneficial for the dataset to be made available to other researchers using a public data repository.

We deeply appreciate the reviewers' positive feedback. Their acknowledgment of the solid methodological approach and the rigorous assessment of genotypic and treatment effects over various developmental stages resonates with our motivation. Their suggestion to make the dataset available in a public data repository for other researchers is well-taken. We are committed to data sharing and we are creating a dedicated platform to facilitate the accessibility of our research data to the scientific community. Given its size and complexity, we currently hold the dataset available upon reasonable request to the corresponding authors.

Weaknesses:There are several primary weaknesses, described below:Sex was not considered in the analyses.The number of experimental animals of each sex are not clearly represented in the paper, but are buried in supplemental tables, and the Ns for the RNAseq are unclear. No analyses were done to examine sex differences in male/female DS or WT animals with or without treatment. Body measurements will greatly vary by sex, but this was not taken into consideration during assessments. As such, there is a high amount of variability within each cohort measured for body assessments (tibia, body weight, skeletal development etc.). Supplemental table 14 had the list of each animal, but not collated by sex, genotype or treatment, making it difficult to assess the strength of each measurement.

Our study primarily concentrated on providing a holistic understanding of the impact of trisomy and GTE-EGCG treatment on Down syndrome, and was not explicitly designed to investigate sexual dimorphism. However, instead of reporting on only one sex and thereby obviating sex as a source of variation, as in previously published studies, we decided to include both male and female mice within the study design to represent a more realistic portrayal of the nature of Down syndrome in a heterogeneous population. By encompassing both sexes, we aim to better capture the variability in Down syndrome.

As we do acknowledge the significance of sex bias in scientific research, we considered performing post-hoc analyses to test the effect of sexual dimorphism, but found that our dataset was underpowered to obtain reliable results, since our experiments were not a priori designed to investigate this question and sample sizes for each sex by separate were not large enough. Nevertheless, considering the reviewer’s comment, we have taken specific steps to improve the representation of sex-related information and to enhance the clarity of our manuscript.

First, we have redesigned all figures using empty and full symbols to distinguish male from female mice within each analysis, providing readers with an immediate sense of the sex distribution in each experimental group. Moreover, we have modified Supplementary Table 1 to offer a comprehensive breakdown of the number of male and female mice for each test, along with their respective genotypes and treatment groups. This table aims to make the sample size and sex distribution within our study as transparent as possible for our readers. While we acknowledge that our study lacked the statistical power to perform a detailed sex-based analysis, the visual representation of sex in our data shows which systems are mainly affected by sexual dysmorphism. This evidence can guide future investigations directly designed to investigate sexual effects in certain systems or structures.

Key results are not clearly depicted in the main figuresRigorous assessment of each figure and the clarity of the figure to convey the results of the analysis needs to be performed. Many of the figures do not clearly represent the findings, with authors heavily relying on supplemental figures to present details to explain results. Figure legends do not adequately describe figures; rather, they are limited to describing how the analysis is performed. For example, LDA plots in Figure 4 do not clearly convey the results of metabolite analysis.Overall, the amount of data presented here is overwhelming, making it difficult to interpret the findings. Some assessments that do not add to the overall paper need to be removed. Clarifying the text, figures and trimming the supplement to represent the data in a manner that is easily understood will improve the readability of the paper. For example, perhaps measures which are not strongly impacted by genotype could be moved to the supplement, because they are not directly relevant to the question of whether GTE-EGCG reverses the impact of trisomy on the measures.

As rightly pointed out by the reviewers, the vast amount of data generated by our holistic and longitudinal approach is one of the primary strengths, but also an important challenge in our study. Our dataset encompasses a comprehensive assessment of the effects of treatment and genotypic alterations at multiple time points during growth and aging. This multi-dimensional evaluation is pivotal to our research, and relegating data to supplementary material would restrict access to this holistic understanding. Our aim is to provide readers with a complete view of the complex interactions we have explored, and retaining this data in the main text is essential to uphold the integrity of our work.

Indeed, we specifically chose to submit or manuscript to eLife because this journal allows to access supplemental information directly from the text and figures in the main manuscript and best aligned with our approach to data presentation. The eLife format permits us to offer readers a quick and informative overview of all the data within the main figures employing multivariate techniques such as Linear Discriminant Analysis or Principal Component Analysis. Subsequently, we supply more detailed analyses in the supplementary figures for readers who wish to delve deeper into specific aspects. Furthermore, while certain figures may be categorized as supplementary, for us it is crucial, and we would like to emphasize, that every result is comprehensively described in the main text.

Acknowledging the concerns raised about the density of our paper and the potential challenges in interpreting the findings, we have conducted a thorough review of the text and figure legends. We have made revisions with the goal to enhance clarity and readability. We have made dedicated efforts to ensure that readers can readily grasp the significance of our results and appreciate the intricacies of our findings. We firmly believe that with these revisions, our chosen approach is the most effective means of presenting the richness of our data and maintaining the integrity of our findings.

Lack of clarity in the behavioral analysesBehavioral assessments are not clearly written in the methods. For example, for the novel object recognition task, it isn't clear how preference was calculated. Is this simply the percent of time spent with the novel object, or is this a relative measure (novel:familiar ratio)? This matters because if it is simply the percent of time, the relevant measure is to compare each group to 50% (the absence of a preference). The key measures for each test need to be readily distinguished from the control measures.There are also many dependent behavioral measures. For example, speed and distance are directly related to each other, but these are typically reported as control measures to help interpret the key measure, which is the anxiety-like behavior. Similarly, some behavioral tests were used to represent multiple behavioral dimensions, such as anxiety and arousal. In general, the measurements of arousal seem atypical (speed and distance are typically reported as control measures, not measures of arousal). Similarly, measures of latency during training would not typically be used as a measure of long-term memory but instead reported as a control measure to show learning occurred. LDA analysis requires independence of the measures, as well as normality. It does not appear that all of the measures fed into this analysis would have met these assumptions, but the methods also do not clearly describe which measures were actually used in the LDA.

We agree with the reviewers’ concerns about the clarity of our behavioral analyses and we have thus added information to the methods section to clarify the procedures. Specifically, for SPSN, social approach was recorded as time spent close to STR1, and a preference ratio was calculated as Pref = 100* Time close to STR1/(Time close to STR1 + Time close to Empty). Social recognition memory was scored as preference towards STR2 and calculated as Pref = 100* (time close to STR2) / (Time close to STR1 + Time close to STR2). For NOR, preference for novel object was calculated as Pref=100* Time novel object / (Time familiar object + novel object).

With regards to the different variables reported for the behavioral protocols, we agree that some measures, such as path length and speed can be used as control measures. For example, in an open field test, path length is an important control measure to assess whether an animal is engaged in the task. However, if an animal is actively moving, the amount of distance covered can but does not have to correlate with the amount of time that a mouse spends in the center of the open field. Using the measure of distance covered as a measure for general arousal and time spent in the center as a measure for anxiety related behavior allows a more nuanced evaluation of animal behavior. For instance, two animals spending similar amounts of time in the center may exhibit differences in the distance they cover. In this scenario, we would argue that anxiety related behavior (defined as exploring the center of an open field) would not reflect well a behavioral difference between the two animals, while the aspect of arousal clearly is a differencing factor.

Regarding the PA task and the use of latency during training, we agree that typically latency during training can be used as control measure to show that learning occurred. However, our study involved testing animals at two distinct time points. Contextual fear conditioning creates very robust memory traces that can persist for weeks or even months, and therefore the starting premise is very different when repeating the test. Initially, the animals were experimentally naïve and had not yet experienced a foot shock, leading to a rapid entry into the dark box. However, after experiencing the first CS-US presentation, a robust and persistent contextual fear memory trace is formed. Therefore, the latency observed in the second training phase of the PA reflects in essence long-term contextual fear memory, that is robustly displayed in WT animals but less in treated WT and TS animals. We have included this clarification in the methods and results sections.

Finally, we want to thank the reviewer for noticing the error in the LDAs, as the analysis was indeed performed including dependent variables for some systems. We have re-evaluated the LDAs for the behavioral tests and tibia microarchitecture tests, excluding dependent variables. As a result, the text and significance levels have been adjusted accordingly. To enhance transparency and clarity, we have included Supplementary Table S21, which precisely outlines the variables included in each LDA.

Unclear value of RNAseqRNAseq was performed in cerebellum, a relatively spared region in DS pathology at an early time point in disease. Further, the expression of 125 genes triplicated in DS was shown in a PCA plot to highly overlap with WT, indicating that there are minimal differences in gene expression in these genes. If these genes are not critical for cerebellar function, perhaps this could account for the lack of differences between WT and Ts65Dn mice. If the authors are interested in performing RNAseq, it would have made more sense to perform this in hippocampus (to compare with metabolites) and to perform more stringent bioinformatic analysis than assessment by PCA of a limited subset of genes. Supplementary Table S14, which shows the differentially expressed genes, appears to be missing from the manuscript and cannot be evaluated. Additionally, the methods of the RNAseq are not sufficiently described and lack critical details. For example, what was the normalization performed, and which groups were compared to identify differentially expressed genes? It would also be worthwhile to describe how animals were identified for RNAseq-were those animals representative of their groups across other measures?

We acknowledge the reviewers' comments on the RNAseq analysis and would like to provide additional insights into our rationale and choices for this analysis. The primary aim of our RNAseq analysis was to offer supplementary evidence in support of the broader context of our paper. Rather than focusing on specific genes, our aim was to assess potential alterations in transcription within genes triplicated in the mouse model and explore differentially expressed genes across the entire genome. Therefore, we conducted a global analysis of the triplicated genes using a PCA and analyzed the differentially expressed genes across the entire genome as shown in Supplementary Table S14. The table was originally included as a separate Excel file but apparently it was not received by the reviewers. We have contacted the eLife editorial to ensure its inclusion in the current version. Furthermore, we have modified the text to clarify that both the triplicated genes and the entire genome were analyzed.

Regarding the use of cerebellum instead of hippocampus, we agree with the reviewers that the hippocampus is a major tissue of interest in the study of Down Syndrome since it mostly relates to cognition. Trisomic patients, however, also display other typical features such as for example a delay in the acquisition of motor skills. Here we decided to focus on the cerebellum as it is primarily associated to the locomotor system but also plays a role in other cognitive functions such as language processing and memory. Furthermore, at the time of the RNAseq analysis, the mice were 8 months old, equivalent to the adult human stage, and previous studies have shown transcriptomic alterations in this tissue and mouse model (Olmos-Serrano et al., 2016; Saran et al., 2003).

The lack of observable differences between WT and Ts65Dn mice in our PCA analysis may be attributed to several factors as discussed in our article. First, the high variability within each group, inherent to the complexity of DS, may obscure inter-group differences. Additionally, the subtlety of gene expression differences between WT and trisomic mice in the set of triplicated genes, as suggested by other transcriptomics studies on DS (Aït Yahya-Graison et al., 2007; Lyle et al., 2004; Olmos-Serrano et al., 2016; Saran et al., 2003), may contribute to the limited distinctions observed. Furthermore, regarding treatment effects, the timing of the RNAseq analysis should be considered since it was conducted at the endpoint, three months after treatment cessation. This temporal aspect could imply that the effects of the drug are not persistent, and a molecular memory might not be formed and maintained.

Nevertheless, we appreciate the reviewers' constructive comments and acknowledge the potential for more stringent bioinformatic analyses. While our intention was to provide an initial, global perspective, we are eager to support further investigations that delve deeper into the complexities of DS-related molecular mechanisms. Consequently, the dataset is available for other researchers to explore more specific questions upon request.

Finally, we have updated the methods section of the article to offer more detailed information on RNAseq processing and analysis. We have also clarified that all the surviving mice were included in the analysis.

**Recommendations for the authors:**
(1) Please add power calculations for each of the assessments.

We would like to clarify that we had already conducted power calculations as part of the initial planning and design phase of our study. After data acquisition and analysis, we have utilized appropriate statistical methods to interpret the results based on the data we have collected. Given that we had conducted a priori power calculations prior to data collection and that our analysis is based on the acquired data, we do not see the added value in including post hoc power calculations. Our primary focus has been on performing the correct statistical analyses to accurately interpret the results and draw meaningful conclusions.

(2) Introduction has some excessive references for each statement, which are not necessary. For instance: lines 67-73 are only references for 1 statement and lines 74-76 are references for a 2nd statement in the same sentence.

We have removed redundant references.

(3) Introduction: Lines 136-146 Gene names need to be spelled out, not just the IDs. Were these studies done in human or mouse models of DS?

We have spelled out the names of the genes.

(4) Why was brain volume and brain structure size normalized to body weight, not clearly explained?

The choice to normalize brain volume and brain structure size to body weight was a deliberate decision made to address potential confounding factors in our study. In the case of trisomic (TS) mice, they are generally smaller in size compared to their wild-type (WT) counterparts. The same may hold true for sex-related size differences. Without normalization, assessing brain volume and structure size could be misleading, as it might reflect the differences in overall body size rather than providing insights into the specific aspects of brain structure that we aimed to investigate. We have clarified this in the methods section.

(5) In cognitive tests, some of the WT data represented in Figure 3 does not match supplemental findings. Again power calculations may indicate a higher number of WT mice are needed to clarify this discrepancy.

We appreciate the reviewers' observation regarding the disparities between the data presented in Figure 3 and the supplemental figures. We would like to clarify that these variations are a result of the distinct analytical approaches employed in the two sets of data.

In Figure 3 and all main figures, the data were analyzed using multivariate tests, which consider multiple variables simultaneously and are particularly suited for investigating the collective impact of multiple factors. Conversely, the results shown in the supplementary figures were derived from univariate tests, which focus on individual variables and are well-suited for addressing specific questions related to each variable in isolation. The discrepancies between the data in the main figures and the supplementary figures can be attributed to the differences in the analytical methods chosen.

As for the suggestion of conducting power calculations to address the observed differences, we believe that the differences in data are inherent to the distinct analytical strategies and the specific research questions each analysis intended to answer. Power calculations may not be the most suitable approach in this context, as they pertain to sample size planning for hypothesis testing and may not reconcile the inherent dissimilarity between multivariate and univariate analyses.

Aït Yahya-Graison, E., Aubert, J., Dauphinot, L., Rivals, I., Prieur, M., Golfier, G., . . . Potier, M. C. (2007). Classification of human chromosome 21 gene-expression variations in Down syndrome: impact on disease phenotypes. Am J Hum Genet, 81(3), 475-491. https://doi.org/10.1086/520000

Lyle, R., Gehrig, C., Neergaard-Henrichsen, C., Deutsch, S., & Antonarakis, S. E. (2004). Gene expression from the aneuploid chromosome in a trisomy mouse model of down syndrome. Genome Res, 14(7), 1268-1274. https://doi.org/10.1101/gr.2090904

Olmos-Serrano, J. L., Kang, H. J., Tyler, W. A., Silbereis, J. C., Cheng, F., Zhu, Y., . . . Sestan, N. (2016). Down Syndrome Developmental Brain Transcriptome Reveals Defective Oligodendrocyte Differentiation and Myelination. Neuron, 89(6), 1208-1222. https://doi.org/10.1016/j.neuron.2016.01.042

Saran, N. G., Pletcher, M. T., Natale, J. E., Cheng, Y., & Reeves, R. H. (2003). Global disruption of the cerebellar transcriptome in a Down syndrome mouse model. Hum Mol Genet, 12(16), 2013-2019. https://doi.org/10.1093/hmg/ddg217